# Mind2Web 2:
# Evaluating Agentic Search with Agent-as-a-Judge

**Boyu Gou**[1][*]   **Zanming Huang**[1][*]   **Yuting Ning**[1][*]   **Yu Gu**[1]   **Michael Lin**[1]
**Weijian Qi**[1]   **Andrei Kopanev**[1]   **Botao Yu**[1]   **Bernal Jiménez Gutiérrez**[1]   **Yiheng Shu**[1]
**Chan Hee Song**[1]   **Jiaman Wu**[1]   **Shijie Chen**[1]   **Hanane Nour Moussa**[1]   **Tianshu Zhang**[1]
**Jian Xie**[1]   **Yifei Li**[1]   **Tianci Xue**[1]   **Zeyi Liao**[1]   **Kai Zhang**[1]   **Boyuan Zheng**[1]
**Zhaowei Cai**[2]   **Viktor Rozgic**[2]   **Morteza Ziyadi**[2]   **Huan Sun**[1]   **Yu Su**[1]

[1]The Ohio State University   [2]Amazon AGI

https://osu-nlp-group.github.io/Mind2Web-2/

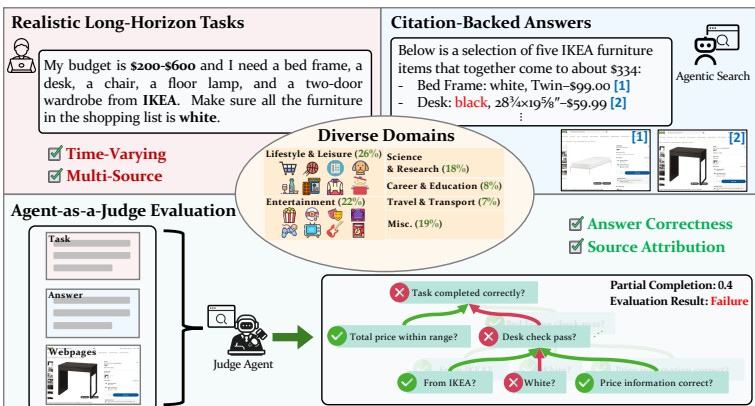

Figure 1: Mind2Web 2 features realistic and diverse long-horizon web search tasks and a novel Agent-as-a-Judge framework to evaluate complex, time-varying, and citation-backed answers.

## Abstract

Agentic search such as Deep Research systems—where agents autonomously browse the web, synthesize information, and return comprehensive citation-backed answers—represents a major shift in how users interact with web-scale information. While promising greater efficiency and cognitive offloading, the growing complexity and open-endedness of agentic search have outpaced existing evaluation benchmarks and methodologies, which largely assume short search horizons and static answers. In this paper, we introduce Mind2Web 2, a benchmark of 130 realistic, high-quality, and long-horizon tasks that require real-time web browsing and extensive information synthesis, constructed with over 1,000 hours of human labor. To address the challenge of evaluating time-varying and complex answers, we propose a novel Agent-as-a-Judge framework. Our method constructs *task-specific judge agents* based on a tree-structured rubric design to automatically assess both answer correctness and source attribution. We conduct a comprehensive evaluation of ten frontier agentic search systems and human performance, along with a detailed error analysis to draw insights for future development. The best-performing system, OpenAI Deep Research, can already achieve 50-70% of human performance while spending half the time, highlighting its great potential. Altogether, Mind2Web 2 provides a rigorous foundation for developing and benchmarking the next generation of agentic search systems.

[*]Equal Contribution. Correspondence: {gou.43, huang.5758, ning.151, sun.397, su.809}@osu.edu

39th Conference on Neural Information Processing Systems (NeurIPS 2025) Track on Datasets and Benchmarks.

# 1   Introduction

Web search has long been the gateway to the world's knowledge, underpinning everything from everyday fact-checking to frontier scientific discovery. The core techniques supporting web search have undergone constant evolution in the past decades, from TF-IDF [32] for term statistics to PageRank [3] for network analysis and learning to rank [4, 20] for supervised learning. Yet the core interaction model has remained essentially unchanged: users issue a query, receive a ranked list of URLs, and must manually open, read, and synthesize multiple webpages to answer complex questions. Current web search is inherently *user-driven*: it retrieves pieces of information but relies on users to interpret and assemble those pieces. This places a significant cognitive load on users, especially as the complexity of the digital world grows.

Recent advances in large language models (LLMs) have sparked the development of *agentic search* systems. Rather than taking keyword queries and returning lists of links, agentic search systems decompose and plan for complex queries, iteratively search the web and interact with dynamic websites, and synthesize information into a citation-backed response. In recent years, agentic search has quickly progressed from *search-augmented LLMs* (e.g., ChatGPT and Perplexity Search) to LLM-based *autonomous web agents* [1, 8, 24, 27, 44, 46] and recent *Deep Research* systems [10, 26] specifically optimized for long-horizon browsing and search behavior. By offloading many low-level tasks, such as query decomposition and reformulation, web browsing, and basic analytics, to a tireless AI agent, agentic search promises to empower human users to focus their cognitive capacity on more important matters like oversight and critical decisions, improving both search efficiency and quality.

However, the rapidly growing complexity of agentic search systems and their tasks is leading to an *evaluation crisis*: how to evaluate the result of a long-horizon task that an AI agent or human produces after taking possibly an hour and hundreds of actions across dozens of websites? Meanwhile, automated and reliable evaluation has proven crucial for the iterative development of AI technologies, especially in the early stages [7, 14, 43]. For agentic search, evaluation is also critical for establishing its *trustworthiness*—while traditional search requires the user to read original documents and verify information, an agent that synthesizes answers must be relied on to be correct and unbiased. Automated evaluation serves as the first line of defense to detect whether an agent is just hallucinating plausible-sounding answers or the cited sources verifiably back them.

Existing benchmarks and evaluation methodologies struggle to keep up with the growing complexity of agentic search. Many benchmarks have been proposed for autonomous web agents [8, 21, 39, 40, 46] but they primarily focus on tasks of a moderate horizon (e.g., up to 10 actions) that can be completed on a single website. Several benchmarks cover cross-website search tasks [23, 34, 41], including most recently BrowseComp [36] from OpenAI. However, to facilitate automated evaluation, a common compromise was made: they focus on tasks with *predefined, time-invariant answers*, oftentimes just a single answer string. While these benchmarks still provide valuable signals for evaluating certain aspects of agentic search systems, they are far from the full spectrum of tasks that current and future systems are facing. Consider an everyday task already within reach of current Deep Research systems, shown in Figure 1. It does not have a predefined answer but requires interacting with live websites to get real-time information. A corresponding agent trajectory may span dozens to hundreds of actions on the IKEA website, let alone more complex tasks that span many websites. We need new evaluation methodologies and benchmarks for such *long-horizon, time-varying* tasks.

In response to these challenges, we propose Mind2Web 2, a new benchmark designed to rigorously evaluate agentic search systems on realistic and long-horizon tasks involving real-time web search and browsing. It consists of 130 high-quality tasks across diverse practical domains. Each task has undergone multiple stages and hours of expert labor for polishing and validation to ensure its realism, complexity, and verifiability. Approximately, at least 1,000 hours of human labor are spent to construct the benchmark, including the tasks and their evaluation scripts.

Agentic search systems typically produce lengthy, time-varying answers (e.g., the product catalog of a shopping website constantly changes) ranging from hundreds to thousands of words on these tasks. The complexity is far beyond what conventional LLM-as-a-Judge [45] methods are used for. Therefore, we propose a novel *Agent-as-a-Judge* framework to automatically yet reliably evaluate such complex answers. The key insight behind our evaluation methodology lies in the *generation-verification asymmetry*: while the generated answers can vary substantially across agents, search strategies, or query times, we know *a priori* what each task is looking for and can design a *task-specific*

*rubric* to specify the evaluation logic. At a high level, a rubric evaluates two main aspects of an answer: *correctness* (i.e., whether the answer satisfies all the requirements of the task) and *attribution* (i.e., whether every statement in the answer can be attributed to the cited sources). At the operational level, a rubric is structured as a tree that breaks down the evaluation into hierarchical evaluation nodes, where each leaf node conforms to a binary judgment and the internal nodes aggregate and propagate the results toward the root following various aggregation logic. Given a task, we develop a *task-specific judge agent*, an agentic workflow interleaving LLM-based information extraction, LLM-as-a-Judge, and tool calls following our unified rubric design, to automatically evaluate complex answers from agentic search systems (see Figure 1 for illustration). Due to the complexity of our tasks, the rubric trees are also highly complex, with an average of 50 nodes and a max of 603 nodes (Table 2 (a)). Yet, rigorous human evaluation of our judge agents shows an evaluation accuracy of 99%, demonstrating their exceptional reliability (§4.4).

We evaluate ten frontier agentic search systems on Mind2Web 2 and also compare them with human performance. Overall, the results show a clear advantage of Deep Research systems over search-augmented LLMs and web agents like Operator, owing to their ability to effectively leverage advanced tools and stay focused over a long horizon. Our results also reveal that current systems still struggle with time-varying tasks that require real-time information and highlight the need for agentic search systems to integrate the ability to interact with live websites. Finally, even though current systems still underperform humans, the best-performing system, OpenAI Deep Research, can already achieve 50-70% of human performance while spending half the time. It also outperforms humans on some tasks requiring great attention to detail and exhaustiveness in the search. After all, humans are subject to cognitive fatigue and a limited working memory. Agentic search presents a substantial potential in augmenting human cognition by automating legwork and allowing us to focus our limited cognitive capacity on things that matter more, such as critical decisions and oversight.

## 2    Related Work

**Agentic Search.** We define *agentic search* as systems that iteratively and autonomously tackle complex search tasks using a combination of tools (e.g., search APIs, retrievers, or web browsing). The autonomy is typically powered by LLMs that decompose the initial search task, dynamically reason and plan based on the accumulating information, or interact with live websites. Early systems like MindSearch [6], ChatGPT and Perplexity Search augment LLMs with search APIs to iteratively search for up-to-date information. However, solely relying on conventional web search inherits its limitations. For example, many websites dynamically render information not indexed by search engines based on user interaction. Autonomous web agents [8, 24, 40, 46], especially those with visual perception of the web [11, 17, 31, 44], have emerged to browse the real-time web as humans do. OpenAI's Operator [27], with specialized reinforcement learning training, represents the current frontier [39]. Recent advances in reasoning models [12, 16] have enabled the development of Deep Research systems [10, 15, 26] that leverage a suite of advanced tools, including search APIs and web browsing, to conduct substantially longer-horizon and deeper research on complex topics. However, there is yet a benchmark designed to simultaneously evaluate this broad spectrum of agentic search systems, a gap that our work aims to bridge.

**Benchmarks and Evaluation Methodologies.** Most existing benchmarks for web agents focus on evaluating whether an agent can autonomously perform certain processes on a single website [8, 13, 17, 21, 39, 40, 46]. The tasks tend to be short (e.g., less than 10 actions) and transactional (e.g., purchasing a flight ticket). Therefore, they can be useful for evaluating the web browsing aspect of agentic search but not the whole system. Several recent benchmarks have a stronger focus on search over the open web [23, 34, 36, 37, 41]. However, for the feasibility of automated evaluation, these benchmarks have made a common compromise: they limit the benchmark to tasks with *predefined, time-invariant answers*, oftentimes just a single answer string. The BrowseComp benchmark [36] from OpenAI, a concurrent work to ours, is representative of this evaluation methodology. Similar to ours, it also leverages the generation-verification asymmetry. It specifically targets tasks that are *hard to solve but easy to verify* (e.g., the answer is often a unique, unambiguous string but may require combing through hundreds of webpages to find it). This strategy is adopted to sidestep the challenge of automatically evaluating complex, time-varying answers, but at the cost of systematically deviating from the true user query distribution. In contrast, we take this challenge head-on with

a novel Agent-as-a-Judge methodology. That allows our benchmark to include more realistic and complex tasks that require a comprehensive answer with real-time information.

LLM-as-a-Judge [45] has been widely used in evaluating complex tasks, including for web agents [13, 28, 39]. However, the complexity of agentic search is far beyond what a few LLM calls can evaluate, necessitating an Agent-as-a-Judge approach [35, 47]. PaperBench [35] (a concurrent work) is most related to ours in that it also adopts a tree-structured rubric, though it is manually written by human experts and used to evaluate the replication of AI research. Our work goes further by largely automating the generation of rubrics. We also have more sophisticated score aggregation methods beyond simple weighted averaging due to the diversity of our tasks. Finally, our attribution evaluation is also related to the attribution literature [9, 18, 19, 42].

# 3 Mind2Web 2

## 3.1 Overview

We introduce Mind2Web 2, a novel benchmark designed to rigorously evaluate agentic search systems on realistic and complex information-gathering tasks involving real-time web search and browsing. There are two main challenges in constructing such a benchmark:

- *How to collect sufficiently complex yet realistic tasks?*
- *How to automatically and reliably evaluate the complex answers generated by different agentic search systems?*

In §3.2, we discuss how we propose, refine, and validate tasks, where we spend hours of expert labor on each task to ensure validity, realism, and verifiability. To tackle the significant evaluation challenge, we propose a novel Agent-as-a-Judge framework that evaluates both the *correctness* (i.e., whether the answer satisfies all the requirements of the task) and *attribution* (i.e., whether each statement in the answer can be attributed to the cited sources). Specifically, we describe our rubric design in §3.3 and the development of judge agents in §3.4, with benchmark statistics in §3.5.

## 3.2 Task Collection

The tasks in Mind2Web 2 shall have the following characteristics: (1) *Realistic and diverse*. Tasks must reflect practical user needs in diverse domains, providing substantial real-world value when solved. (2) *Long-horizon and laborious*. Tasks require substantial human effort due to an extended length and breadth of the required searches. (3) *Objective and verifiable*. Each task must have clearly defined evaluation criteria that are verifiable by checking the answer text in addition to the cited source webpages. (4) *Time-varying*. We encourage time-varying tasks with answers that could change over time, although it is not a requirement for every task.

Our task collection team consists of three groups of annotators (all are experienced computer science students or professionals): *task proposers*, *refinement experts*, and *validation experts*, who lead different stages of the procedure. First, *task proposers* freely generate task ideas based on their authentic search needs or inspirations from our provided domain guidelines, ensuring initial alignment with the realism and laboriousness desiderata. Second, trained *refinement experts*, collaborating closely with the task proposers, iteratively revise or filter tasks to enforce strong alignment to our task principles. Finally, experienced *validation experts* manually attempt and verify each refined task, focusing on task feasibility, potential subtle issues, and practicability of the evaluation. Only tasks independently validated by at least two validation experts are included in Mind2Web 2.

## 3.3 Rubric Tree

To support reliable, scalable and automated evaluation of the tasks in Mind2Web 2, we design a unified tree-structured rubric formulation. Each leaf node represents a criterion that can be assessed through straightforward verification, yielding a binary score of $0$ or $1$. These binary scores are then aggregated iteratively by parent nodes to determine the scores for higher-level criteria.

Specifically, a rubric may include two types of nodes. Each node is either a *critical node*, representing an essential criterion whose failure immediately fails its parent node (e.g., the budget evaluation node *(a)* or any child node of *(b)* in Figure 2), or a *non-critical node*, allowing partial scoring at its parent

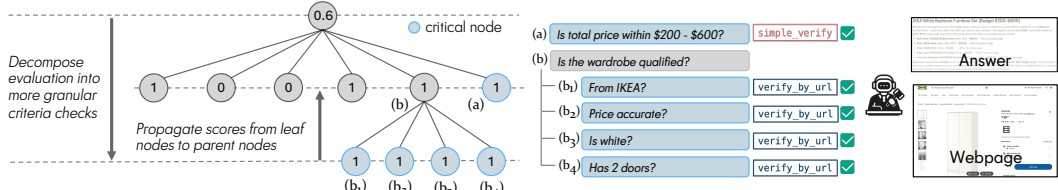

Figure 2: Example of tree-structured rubrics. Top-down, task goals are decomposed into a tree structure; bottom-up, binary scores from leaf nodes are aggregated into the overall task score. The leaf nodes are verification of low-level criteria, implemented by various functions of judge agents (e.g., simple_verify: verify a simple factual or logical statement; verify_by_url: verify whether a statement in the answer is backed by a cited webpage). See more discussion in §3.3 and §3.4.

node (e.g., we independently assess each of the five requested furniture and give partial credit in Figure 2). Additionally, some nodes may be marked as *sequential*, reflecting a logical dependency among their child nodes, where a failure at an earlier node *short-circuits* all subsequent nodes. For example, if a task requires finding a certain paper and subsequently the email of its first author, failing to find the correct paper makes it pointless to evaluate the email node.[2]

Intuitively, the score aggregation employs a *gate-then-average* strategy: critical nodes serve as gating conditions when paired with non-critical nodes. In practice, critical nodes often represent basic and essential constraints rather than incremental progress, thus their scores do not directly contribute to the averaging process for partial scoring, but instead function solely to warrant the meaningfulness of aggregating scores from non-critical nodes. Finally, if a node only contains critical child nodes, which indicates that each child represents a necessary condition for the parent criterion, the score of the parent node directly depends on the passing of all these critical child nodes (e.g., in Figure 2, the wardrobe node *(b)* gets a score 1 only if all the child nodes pass; otherwise 0).

Formally, let $v$ be a node in the rubric tree with child nodes $C(v)$. We partition child nodes into critical nodes $K(v) \subseteq C(v)$ and non-critical nodes $N(v) = C(v) \setminus K(v)$. The score $s(v) \in [0, 1]$ of $v$ is recursively defined as:

$$
s(v) = \begin{cases}
0, & \text{if } \exists u \in K(v), \ s(u) < 1, \\
\frac{1}{|N(v)|} \sum_{u \in N(v)} s(u), & \text{if } \forall u \in K(v), \ s(u) = 1 \text{ and } |N(v)| > 0, \\
1, & \text{otherwise.}
\end{cases}
$$

We define two metrics based on the final aggregated score at the root node: (1) **Partial Completion**, the average root node scores across all tasks, reflecting the partial satisfaction based on the fine-grained evaluation, and (2) **Success Rate**, the percentage of tasks achieving a perfect root node score of 1, indicating full task completion with all criteria satisfied.

### 3.4 Rubric-based Judge Agent

Following the rubric design in §3.3, each task in Mind2Web 2 is evaluated by a dedicated *judge agent*, which is a task-specific agentic workflow that implements the rubric-style evaluation wrapped in a Python script. A judge agent takes the answer text (including the source citations) as input, evaluates each fine-grained criterion (i.e., the leaf nodes of the rubric tree), and calculates the final score by aggregating scores upwards to the root node.

The judge agents primarily leverage two LLM-based tools: (1) *Extractor* that parses answer text to extract structured information (e.g., item names, prices, and URLs), and (2) *Verifier* that applies verification.[3] Take the leaf node *(b₃)* in Figure 2 as an example, the Extractor extracts the corresponding bits of information from the answer, and the Verifier examines the extracted text and the screenshot of the corresponding webpage to determine if the statement is indeed true.

---

[2]This sequential logic is sufficient for our current tasks, though future work can explore other logic.
[3]We use OpenAI o4-mini as the LLM in both tools.

Table 1: Comparison with existing benchmarks for web browsing or search on live websites. **Horizon**: the average number of required actions per task, grouped into Short ($< 10$), Medium ($10-50$), Long ($> 50$). **Time-Varying**: whether the answer can change over time.

| | Horizon | # of Tasks | Time-Varying | Evaluation |
|---|---|---|---|---|
| Online-Mind2Web [39] | Short | 300 | ✓ | LLM-as-a-Judge |
| WebVoyager [13] | Short | 643 | ✓ | LLM-as-a-Judge |
| Mind2Web-Live [29] | Short | 542 | ✓ | Rule |
| BEARCUBS [34] | Short | 111 | ✗ | Answer Match |
| WebWalkerQA [37] | Short | 680 | ✗ | Answer Match |
| GAIA [23] | Medium | 466 | ✗ | Answer Match |
| AssistantBench [41] | Medium | 214 | ✗ | Answer Match |
| BrowseComp [36] | Long | 1,266 | ✗ | Answer Match |
| Mind2Web 2 | Long | 130 | ✓ | Agent-as-a-Judge |

Manually crafting such judge-agent scripts from scratch is prohibitively demanding due to the complexity and granularity of the evaluation criteria. Thus, we first develop a modular Python toolkit encapsulating reusable rubric-management utilities and standardized *Extractor* and *Verifier* modules. This toolkit substantially reduces coding overhead, allowing annotators to focus primarily on rubric design rather than code details. Nonetheless, script creation remains demanding even with this toolkit. To further facilitate the development, we build an LLM-based agentic code generation pipeline that produces an initial version of the scripts. The generated scripts undergo iterative autonomous refinements (including self-debug [5] and self-reflection [22, 33]) to auto-correct minor or common errors. Finally, scripts are rigorously validated through a two-stage human refinement process, which ensures correctness and enhances generalizability across all possible answers. We also conduct a human evaluation of our rubrics and judge agents in §4.4. Further details about rubrics and script development are provided in Appendix D. An exemplar script is provided in Appendix G.

## 3.5 Benchmark Statistics

Through the pipeline described in §3.2-§3.4, we collect a total of 130 carefully curated tasks, each accompanied by a carefully developed judge-agent script. Task distribution across domains is shown in Figure 1 and Appendix C.1. In total, the construction of this benchmark (including both task collection and judge-agent development) involves at least 1,000 hours of human labor.

The statistics of the rubric trees in Table 2 (a) show the complexity of our tasks, with rubric trees having up to 6 layers and 603 evaluation nodes. To further quantify the complexity of our benchmark, we conduct a human performance study on a randomly selected subset of 30 tasks (Subset-30). Seven participants are asked to manually complete these tasks (each task by three different participants), allowing us to observe human behaviors and measure human effort associated with the tasks. Results in Table 2 (b) show that our tasks are indeed highly time-consuming for humans: It can take up to one hour and humans need to visit as many as 31 websites and 375 webpages to get the answer. Note that these numbers are underestimated, as participants may make mistakes or omit steps, and are allowed to stop after one hour or if unable to find clear paths to complete the task.

Table 2: Benchmark statistics.

(a) Rubric complexity.

| | Avg | Min | Max |
|---|---|---|---|
| # Leaf nodes | 34 | 3 | 357 |
| # Total nodes | 50 | 4 | 603 |
| Depth | 4 | 2 | 6 |

(b) Human effort required per task (Subset-30).

| | Avg | Min | Max |
|---|---|---|---|
| Time (min) | 18 | 8 | 44 |
| # Websites | 8 | 3 | 31 |
| # Webpages | 110 | 38 | 375 |

Table 1 shows the comparison of Mind2Web 2 to other related benchmarks. As discussed in §2, Mind2Web 2 is the only agentic search benchmark to date focusing on long-horizon, time-varying tasks, and is made possible due to our advanced Agent-as-a-Judge evaluation methodology. It is worth noting that even though there are only 130 tasks, each task contains dozens to hundreds of fine-grained evaluation nodes, thus still providing sufficient differentiation power.

We split our benchmark into a *development set* (10 tasks), and a *test set* (120 tasks), and release all the tasks and evaluation scripts. We will maintain a leaderboard for the test set.

Table 3: Main evaluation results. We report the partial completion score, full-task success rate, Pass@3, average time (in minutes), average answer length (in words), and their standard deviation. *: To reduce human workload, the human study is conducted on Subset-30 as described in §3.5.

| | Partial Completion | Success Rate | Pass@3 | Time (min) | Answer Length |
|---|---|---|---|---|---|
| ChatGPT Search | $0.26_{\pm 0.01}$ | $0.06_{\pm 0.01}$ | 0.11 | $< 1$ | $314_{\pm 4}$ |
| Perplexity Pro Search | $0.28_{\pm 0.02}$ | $0.08_{\pm 0.01}$ | 0.12 | $< 1$ | $408_{\pm 13}$ |
| OpenAI Operator | $0.26_{\pm 0.01}$ | $0.10_{\pm 0.01}$ | 0.17 | $9.74_{\pm 0.21}$ | $160_{\pm 1}$ |
| HF Open Deep Research | $0.26_{\pm 0.01}$ | $0.11_{\pm 0.01}$ | 0.18 | $13.65_{\pm 0.07}$ | $209_{\pm 3}$ |
| Claude Research | $0.32_{\pm 0.03}$ | $0.10_{\pm 0.03}$ | 0.19 | $7.39_{\pm 0.14}$ | $742_{\pm 1}$ |
| Grok DeepSearch | $0.40_{\pm 0.04}$ | $0.18_{\pm 0.02}$ | 0.36 | $2.58_{\pm 0.14}$ | $1,428_{\pm 16}$ |
| Perplexity Deep Research | $0.42_{\pm 0.03}$ | $0.15_{\pm 0.03}$ | 0.26 | $5.67_{\pm 0.13}$ | $585_{\pm 13}$ |
| Gemini Deep Research | $0.45_{\pm 0.03}$ | $0.18_{\pm 0.02}$ | 0.30 | $7.38_{\pm 0.58}$ | $3,357_{\pm 49}$ |
| Grok DeeperSearch | $\underline{0.52}_{\pm 0.02}$ | $\underline{0.27}_{\pm 0.03}$ | **0.40** | $5.72_{\pm 0.27}$ | $1,362_{\pm 24}$ |
| OpenAI Deep Research | $\mathbf{0.54}_{\pm 0.04}$ | $\mathbf{0.28}_{\pm 0.04}$ | **0.40** | $8.40_{\pm 0.71}$ | $559_{\pm 19}$ |
| Human* | $0.79_{\pm 0.01}$ | $0.54_{\pm 0.07}$ | 0.83 | $18.40_{\pm 1.61}$ | $186_{\pm 27}$ |

# 4 Experiments

## 4.1 Experimental Setup

We evaluate agentic search systems of various types on Mind2Web 2. Given the complexity of our tasks, we focus on frontier systems capable of yielding meaningful results, namely, those exhibit sufficient long-horizon search capability and can consistently provide source attributions. We report two primary metrics: **Partial Completion** and **Success Rate**, as defined in §3.3. We run and evaluate each system independently over three runs per task, and we present the averaged metrics along with their standard deviations. Additionally, we introduce **Pass@3**, indicating whether at least one of the three attempts for a task is successful. To further contextualize system performance, we also report behavioral aspects influencing user experience, including the average task completion time and average answer length.[4] We report results on the test set, reserving the public development set for unrestricted exploration.[5]

We include two prominent commercial search products, ChatGPT Search [25] and Perplexity Pro Search [30], which augment LLMs with search capabilities, delivering rapid responses with a limited number of agentic search steps. Additionally, we evaluate a suite of Deep Research systems [2, 10, 15, 26, 30, 38], which are explicitly optimized for extensive information gathering and comprehensive report generation, many of which can sustain continuous running for extended periods (e.g., beyond 30 minutes per query). Lastly, we assess OpenAI Operator [27], one of the most advanced web agents currently available, which performs tasks through direct browser interactions. Hugging Face Open Deep Research [15] is the only open-source system that we find to yield reasonable results at the time of this evaluation; all the other sufficiently capable systems are closed-source.

To provide deeper insights into the practical values of these systems, we further include a human performance study (previously detailed in §3.5), wherein human participants undertake tasks in Subset-30 under fair settings (further elaborated in Appendix E.3).

## 4.2 Main Results

As shown in Table 3, while most tasks in Mind2Web 2 are conceptually straightforward, their tedious nature poses substantial challenges not only for the agent systems but also for human participants, resulting in low success rates (up to 28% for agents and 54% for humans). Moreover, the substantial gap between partial completions and success rates highlights that current systems often demonstrate initial competence but struggle to fully complete tasks accurately.

**Comparison Between Agent Types.** Unsurprisingly, ChatGPT Search and Perplexity Pro Search emerge as the weakest systems, primarily limited by their restricted search horizon and relatively

---

[4]We use the self-reported completion time whenever available; otherwise, we manually record the completion time. Manual recording is limited to Subset-30 to reduce human workload.

[5]Note that Subset-30 is guaranteed to be a subset of the test set.

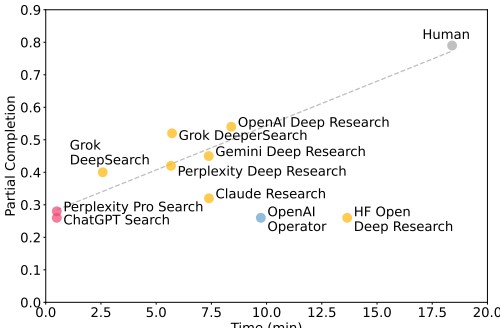
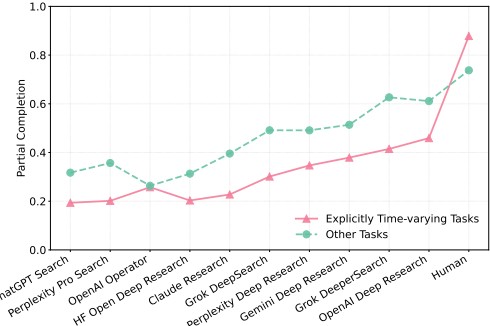

Figure 3: Average Partial Completion against average task completion time.

Figure 4: Average Partial Completion on explicitly time-varying tasks compared to other tasks.

shallow information synthesis abilities inherent to LLMs. In contrast, most Deep Research systems achieve superior performance. These systems are explicitly designed, trained, or prompted for extensive information gathering and sophisticated synthesis tasks, enabling sustained, detailed task engagement. Additionally, several Deep Research systems integrate capabilities of text-only or multimodal web browsing (clicking, scrolling), alongside coding tools (e.g., dedicated virtual environments, Python interpreters), enabling real-time search on live websites as well as advanced reasoning and information synthesis. Operator exhibits notably lower performance compared to Deep Research systems. Compared to agents primarily leveraging search APIs, web agents navigate more complex and noisier environments, manage complex action spaces, and handle substantially longer and more intricate context. These pose substantial challenges to robust long-term reasoning, planning, and memory management, and these challenges are especially amplified and highlighted by the extensive, long-horizon tasks included in Mind2Web 2. Moreover, unlike web agents that sequentially interact with browsers, recent search agents have begun leveraging parallelized retrieval strategies, offering clear advantages in locating information from the vast online content landscape.

**Test-Time Scaling.** As illustrated in Figure 3, we observe clear performance improvements resulting from increased inference time. The benefit is especially evident when comparing systems within the same family (e.g., Grok and Perplexity), given that they presumably share the same underlying models. This observation aligns intuitively with the complexity of our tasks, which inherently demands prolonged searches and sophisticated synthesis: extending inference time enables agents to more thoroughly retrieve, process, and integrate the necessary information. Additionally, performing multiple independent trials for each system substantially enhances the likelihood of task success, as indicated by the improved Pass@3 scores. This further underscores the potential of current agentic search systems to benefit from increased computational resources and inference attempts.

**Struggle with Time-Varying Tasks.** We hypothesize that agentic search systems equipped with no or only limited browsing features might perform worse on time-varying tasks compared to time-invariant tasks. Many of those tasks inherently require live web interactions, for instance, verifying hotel room availability on a specific date. Without real-time browsing, agents often provide outdated or hallucinated information. We identify 57 tasks that are explicitly time-varying (i.e., tasks explicitly associated with relative dates/times, or requiring information like product prices that frequently changes over time). As shown in Figure 4, most of the evaluated systems perform worse on this subset than on the remaining tasks, which supports our hypothesis. Interestingly, OpenAI Operator and human participants, both excelling at interacting with live websites, achieve relatively on-par or superior performance on time-varying tasks. In addition to real-time information, some tasks, such as those requiring advanced filters or visual understanding, also favor browser interaction over search APIs. These collectively highlight the importance of integrating web browsing into agentic search systems, likely contributing substantially to OpenAI Deep Research's superior performance over the other Deep Research systems.

**Promises of Agentic Search.** Despite current limitations, our evaluation already demonstrates early promise of agentic search systems. The best-performing system, OpenAI Deep Research, already achieves 50-70% of human performance while spending less than half the time. Humans are not perfect at many of such complex tasks because we are subject to cognitive fatigue and limited working

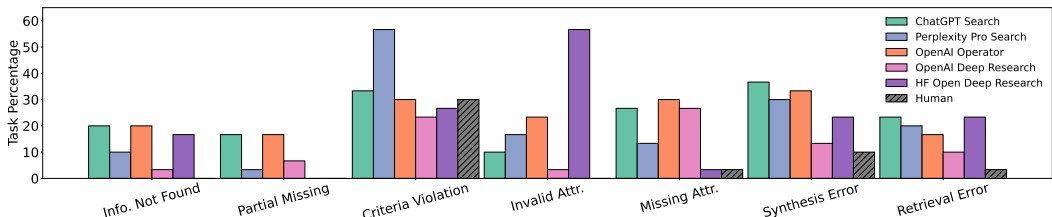

Figure 5: Errors across agents and humans. The bars indicate the percentage of tasks exhibiting each error type. We include results from five agentic search systems and humans.

memory. For instance, in a task that requires retrieval of news articles with nuanced constraints, all the human participants exhibit various forms of oversight or carelessness regarding subtle details or overall task requirements, resulting in task failures. In contrast, most agent systems accurately interpret the task and articles and achieve better performance. Agentic search has substantial potential to augment human cognition by automating away the legwork and allowing us to focus our limited cognitive capacity on things that matter more, such as critical decisions and active oversight.

## 4.3 Error Analysis

We conduct a detailed error analysis over the evaluated systems to gain insights for future development. Specifically, we ask human annotators to manually label error types in the answers for Subset-30, and define seven common and easy-to-identify error categories on *correctness* and *attribution*. Results are shown in Figure 5, noting that a single answer may contain multiple error types. Detailed definitions and examples are provided in Appendix F.1. We also provide additional case studies in Appendix F.2.

**Incompleteness**: We observe a notable gap between human and agent performance regarding task completion. We further divide the errors into two subtypes: (1) *Info. Not Found* (Ex. F.2), i.e., the agent explicitly states failure in retrieving the requested information. (2) *Partial Missing* (Ex. F.3), i.e., the agent provides fewer items or fewer procedural steps than explicitly required by the task. On the one hand, systems not optimized for 'Deep Research' often exhibit early termination behaviors, showing a degree of *laziness*. On the other hand, they inherently lack sufficient capabilities to complete these long-horizon tasks. For example, ChatGPT Search executes only limited search steps and applies simple information synthesis by LLMs without using advanced tools (e.g., Python interpreters in many Deep Research systems), making it challenging for it to find and integrate all necessary information. Systems such as HF Open Deep Research and Operator frequently exhibit total failures at some tasks. Upon closer examination, we find many issues of HF Open Deep Research regarding system errors (e.g., failures in adhering to system prompts to generate valid code to invoke the search tool), which may also apply to other open-source agent systems that simply leverage off-the-shelf models to build deep research systems without post-training the underlying models.

**Criteria Violation** (Ex. F.4): We identify explicit violations of task criteria or factually wrong statements directly identifiable from the answer text. Such errors are prevalent among all the evaluated systems, including humans. Notably, this is the most common error type for humans, primarily due to the tedious and demanding nature of the tasks, where humans appear to struggle to remain patient and careful. For instance, one annotator mistakenly lists the University of Waterloo as a U.S. institution. Interestingly, Deep Research systems (e.g., OpenAI Deep Research) have already surpassed humans in this regard, as they are designed to perform exhaustive searches and analyses to meet users' requirements.

**Invalid Attribution** (Ex. F.5): We often observe expired and fabricated URLs from the answers. One potential reason could be that agents generate URLs directly without actually accessing the webpages. For instance, in a task requiring an Amazon purchase link, HF Open Deep Research directly fabricates a link without accessing Amazon. Surprisingly, Operator also has a high percentage of this error type, even though it actually accesses websites as humans do. From the trajectories, we find that it often mistakenly reports incorrect URLs in its final responses even though it has successfully accessed the correct webpages, which may be partially due to the challenge of generating answers grounded in a long context. For example, in a fellowship identification task, Operator navigates correctly to the correct page but ultimately reports a link that differs by a few words from the correct link.

**Missing Attribution** (Ex. F.6): Claims made in the responses often lack source attribution. Web agents are usually designed or trained on web navigation or citation-free information-seeking tasks. Therefore, in contrast to AI search systems, Operator struggles to follow our instructions to provide attribution. Moreover, LLMs with massive parametric memory sometimes tend to directly produce or hallucinate information without conducting actual searches, even though most of our tasks do require them to search online in order to provide up-to-date information and attribution.

**Unsupported Answer**: The information in the answer may differ from the sources even when valid attribution is provided. We further divide this issue into two subtypes: (1) *Synthesis Error* (Ex. F.8), i.e., the agent synthesizes information incorrectly from correct webpages (e.g., distorting the price listed on a product page). (2) *Retrieval Error* (Ex. F.7), i.e., the provided source is totally irrelevant. Synthesis errors are pronounced in ChatGPT Search and Perplexity Pro Search, which struggle to accurately synthesize from extensive sources without advanced tools (e.g., Python interpreters). Humans also sometimes commit synthesis errors due to carelessness when overwhelmed by large volumes of information. Retrieval errors can result from failing to retrieve relevant information. For example, an agent may retrieve webpages similar but not precisely aligned with the task requirements, subsequently causing the agent to hallucinate seemingly relevant but unsupported details.

### 4.4 Human Evaluation of Judge Agents

Empirically, we have validated the reliability of our judge agents through validation processes. Nonetheless, it remains possible that some evaluation inaccuracies persist. Therefore, to rigorously assess the reliability of our judge agents, we conduct an additional human evaluation. Specifically, this evaluation comprises three phases on 15 randomly sampled tasks, each with evaluation results of two held-out answers from different agent systems. Further details are provided in Appendix D.5, with an additional ablation study of the base model for the judge agents in Appendix D.6.

In the **Rubric-Level Assessment**, the evaluator assesses the overall rubrics of judge agents independently (without viewing answers or automated evaluation results), rating their validity and comprehensiveness. In the subsequent **Node-Level Assessment**, the evaluator acts as the Verifier, manually annotating the leaf-node binary judgments, which are then compared against automated judge-agent results to identify discrepancies. To avoid human annotation mistakes and further confirm the accuracy, we subsequently perform a **Validation of Human Annotation**, wherein an experienced judge-agent developer examines all identified node discrepancies.

**Results and Analysis.** The evaluator fully agrees with all 15 rubrics, offering minor suggestions on the strictness of partial scoring for two rubrics while acknowledging that the existing partial scoring remains reasonable. At the leaf-node level, we identify a total of 35 discrepancies out of 720 verifications. Upon further validation, we discover that 27 of the discrepancies arise from human evaluator errors; the original judgments are correct. This highlights the high complexity and cognitive demand involved in accurately evaluating claims within lengthy answers from agentic search, and reaffirms the reliability of our automated judge agents relative to even well-informed human evaluators. A detailed analysis of the remaining discrepancies is provided in Appendix D.5.

Excluding human mistakes, only 7 out of 720 nodes reflect actual errors, indicating an exceptional accuracy of 99.03%. This demonstrates remarkable reliability, particularly when compared to recent automated evaluation approaches for relatively simpler web tasks [39], where the reported accuracy of the automated evaluation methods typically falls below 90%. We attribute this success to our tree-structured rubric design that cleanly decomposes the complex evaluation, the agentic code generation pipeline for generating judge agents, as well as the rigorous human refinement process.

## 5 Conclusions

In this work, we introduced Mind2Web 2, a novel benchmark specifically designed for comprehensively evaluating agentic search systems on long-horizon information-gathering tasks and time-varying answers. We proposed a scalable, automated, and reliable evaluation framework based on Agents-as-a-Judge that systematically assesses agent performance on open-ended long-horizon search tasks. Our comprehensive empirical analysis, spanning AI-based search engines, deep research systems, and web agents, reveals both their potential and current limitations. Mind2Web 2 serves as a valuable resource and rigorous assessment platform for better advancing agentic search systems.

## Acknowledgments

The authors would like to thank colleagues from the OSU NLP group and Amazon AGI for constructive discussions and generous help, Zishuo Zheng for his exploration of developing long-horizon agentic search agents, Akshay Anand and Scott Salisbury for their help on benchmark construction, the Hugging Face team (Amir Mahla, Aymeric Roucher, Aksel Joonas Reedi, and Thomas Wolf) for their assistance with the evaluation of Hugging Face Open Deep Research as well as covering the inference costs, the Grok team (Piaoyang Cui, Hexiang Hu) for their assistance with the evaluation of Grok DeepSearch and DeeperSearch, and the Amazon AGI team for their valuable feedback and contribution to task collection. This research is sponsored in part by a gift from Amazon, ARL W911NF2220144, NSF CAREER #1942980, and NSF OAC 2112606. The views and conclusions contained herein are those of the authors and should not be interpreted as representing the official policies, either expressed or implied, of the U.S. government. The U.S. government is authorized to reproduce and distribute reprints for government purposes notwithstanding any copyright notice herein.

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

# Table of Contents in Appendix

# A  Limitations

We acknowledge and discuss several limitations in our benchmark design and evaluation methodology:

**Task Coverage and Scope.** While Mind2Web 2 comprises 130 carefully curated tasks spanning diverse practical domains, it cannot encompass all possible real-world information-seeking scenarios. Certain task categories (e.g., vague or highly subjective queries) are excluded due to our focus on realistic, tedious information-gathering tasks and practical considerations for evaluation. Nevertheless, the extensive diversity of included domains, websites, and realistic scenarios still ensures reasonable coverage. Thus, we believe these exclusions do not significantly diminish the benchmark's utility for evaluating and advancing agentic search systems.

**Evaluation Framework Assumptions.** Our evaluation framework relies on URL-based attribution, presupposing that cited URLs provide truthful and credible information, despite potential misinformation on the web. Evaluating the credibility and truthfulness of individual sources is beyond the scope of this work. Additionally, our evaluation and task design assume critical information can be attributed to individual webpages, which may not always hold true for all possible tasks. However, this constraint has not prevented us from developing a large, diverse, and meaningful benchmark.

**Reliance on LLM-based Judgments.** Our evaluation employs LLM-based extractions and verifications. While powerful, LLMs may occasionally introduce extraction errors or incorrect judgments. Empirically, we find the base model (OpenAI `o4-mini`) sufficiently capable for the extraction and verification tasks in this benchmark. Moreover, to mitigate potential inaccuracies and maintain evaluation reliability, we employ multi-stage validation processes, including rigorous human validation and refinements of evaluation scripts. We further conduct human evaluations of judge-agent outputs (Appendix D.5) as well as an ablation study regarding the base model (Appendix D.6), systematically assessing and confirming the overall reliability of LLM-based judgments.

**Limited Analysis on Black-Box Systems.** Our benchmark primarily evaluates state-of-the-art commercial and research-grade agentic search systems. To ensure informative comparisons, we exclude weak systems incapable of meaningful performance, consequently focusing mainly on proprietary or closed-source solutions. This limits our ability to fully interpret performance differences or estimate precise inference costs (e.g., token usage). Nonetheless, our answer-based evaluation framework effectively assesses the capabilities and common failure modes (e.g., pervasive hallucinations) of these black-box systems, offering valuable insights. To partially compensate for limited access, we report metrics such as task completion time and generated answer length, providing relative references for practical efficiency.

# B  Broader Impacts and Ethical Considerations

In this section, we discuss broader impacts from two interconnected perspectives: the broader implications of agentic search systems, and the impacts associated with the release and use of the Mind2Web 2 benchmark.

**Agentic Search Systems.** Advanced agentic search systems promise a transformation in how users interact with the web, shifting from manual, multi-step information gathering to streamlined, automated information synthesis. This change could significantly reduce cognitive load, improve efficiency, democratize sophisticated search capabilities, and support informed decision-making across diverse fields including education, healthcare, commerce, and policy-making.

Despite benefits, enhanced agentic search systems may exacerbate misinformation by generating seemingly credible yet incorrect or unsupported information. Malicious actors could exploit such systems for large-scale disinformation or unauthorized data extraction. Additionally, agentic systems risk perpetuating existing biases found in web content, raising fairness concerns and potentially leading to discriminatory outcomes without careful oversight and transparency. Reliable and scalable evaluation serves as the first line of defense to detect and mitigate such issues.

**Mind2Web 2 Benchmark.** By emphasizing rigorous evaluation through structured rubrics and explicit verification of source attribution, Mind2Web 2 facilitates the development of transparent and accountable agentic search systems. Establishing standardized, robust evaluation practices helps accelerate trustworthy system development and promotes clarity in capability assessments across the research and industry communities.

However, wide adoption of our rubric-based evaluation could lead to automated mass production of training data via reinforcement learning, particularly by resourceful organizations. While this may improve agent capabilities, it also risks overfitting to benchmark-specific tasks and amplifying biases inherent in rubrics or evaluation methods. Consequently, agents might perform poorly in broader, unstructured real-world scenarios or inadvertently introduce systematic biases. We will carefully monitor this potential issue and update the benchmark when necessary.

# C   Details of Task Collection

## C.1   Domain Distribution

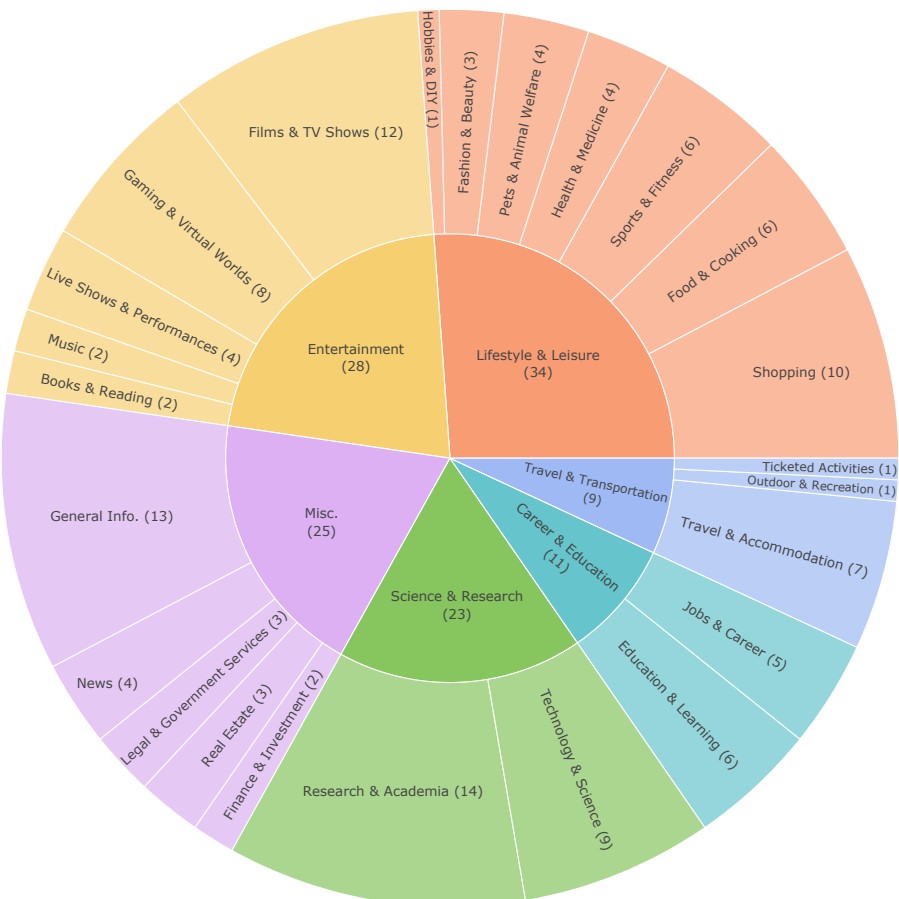

Figure C.1: Mind2Web 2 contains 130 diverse tasks covering 6 broad domains and 24 sub-domains.

During task collection, proposers are provided an initial set of fine-grained domains derived from prior work [8] and further expanded using GPT-4o. Proposers categorize each new task into the most suitable domain, adding new domains as needed. In the subsequent refinement and validation stages, domain assignments are reviewed and adjusted by expert annotators to ensure accuracy and minimize redundancy. Finally, after collecting all 130 tasks, we further refine and consolidate domain categorizations to minimize overlap and redundancy, resulting in the final domain structure presented in Figure C.1.

## C.2   Design Principles of Tasks

To ensure tasks align with the goals of our benchmark and are compatible with our rubric-based evaluation framework, we define and follow these task-design principles:

**Realism.** Tasks should represent authentic and practical user needs. Each task must have clear real-world applicability, avoiding artificial combinations of unrelated steps just for complexity or to challenge AI systems.

**Tediousness (Long-Horizon).** Tasks must require sustained effort due to extensive web search, exploration, and information synthesis. Simple tasks solvable within a few queries are explicitly avoided. Human annotators validate tediousness by confirming each task requires at least five minutes of human effort. Note that it is just the minimum; most tasks in Mind2Web 2 take humans much longer to complete (see statistics in Table 2).

**Clarity and Objectivity.** Task descriptions must be explicit, precise, grammatically correct, and unambiguous. Answer criteria must be clearly stated, avoiding vague or subjective terms (e.g., "*good*," "*effective*," or "*better*"). When domain-specific knowledge is required, it must be clearly defined or explained in the task description. To ensure clarity, tasks undergo ambiguity checks via both manual and LLM-assisted inspection.

**Verifiability.** Tasks must have clearly defined and practically verifiable criteria. The criteria should be verifiable primarily through the answer text itself as well as the expected URL-based provenance. Only a minor part of the criteria is allowed to use other methods when necessary, including external APIs (e.g., Google Maps for distance measurement) and fixed ground-truth answers (or ground-truth answers from fixed URLs).

**Additional Constraints and Exclusions.** To ensure practicality and our focus on web search instead of other intelligent capabilities as well as the reliability of evaluation, the following constraints apply:

- Tasks involving video understanding or non-English websites are excluded from this study.

- Tasks *explicitly* requiring complex reasoning (e.g., summarize a complex research paper) or external tools (e.g., Python interpreters or calculators) are avoided. However, we do not constrain the evaluated systems on how they complete the tasks. They can use whatever tools deemed necessary or helpful.

- Tasks whose answers constantly change (e.g., currency exchange rates which change within a very short period) are excluded to ensure stable evaluation.

- Tasks should avoid reliance on global or overly general qualifiers (e.g., "cheapest," "list all," or "top-$k$") unless these conditions are verifiable (e.g., by a fixed set of URL sources or fixed ground-truth answers).

- We currently assume each verification of attribution can be conducted on a single webpage. Tasks requiring simultaneous verification across multiple webpages, where verification cannot be decomposed into independent single-page validations, are beyond the scope of this benchmark.

These principles are documented and illustrated with concrete examples, serving as guidelines for human annotators. Detailed instructions are provided in Appendix H. Each task is carefully validated and iteratively refined by initial proposers, refinement experts and validation experts to ensure full compliance before final inclusion into Mind2Web 2.

### C.3 Task Collection Pipeline

We collect and refine tasks for Mind2Web 2 under a three-stage pipeline: *Proposal*, *Refinement*, and *Validation*, ensuring adherence to the task principles as well as evaluation practicability. We provide the full instructions of each stage in Appendix H.1

**Task Proposal.** 21 initial task proposers independently generate task ideas aligned with the defined principles. At this stage, proposers conduct self-checks covering major task principles (e.g., realism, tediousness, clarity, verifiability) as well as minor aspects such as grammatical correctness and clarity. Proposers also provide initial draft answers or relevant URLs to facilitate the following refinement and validation phases. A total of 430 tasks are proposed at this stage (excluding incomplete or clearly unsuitable tasks considered as random drafts)

**Task Refinement.** 7 refinement experts further review and iteratively refine each proposed task together with the initial proposers. During refinement, the experts carefully evaluate tasks for practicality, clarity, and adherence to the defined principles, suggesting necessary adjustments to task descriptions, verification criteria, or expected answers. The refinement ensures that tasks remain

realistic and challenging yet clearly defined and objectively verifiable. A total of 155 tasks are retained at this stage.

**Task Validation.** Finally, each task undergoes validation by another two or more independent annotators, where 4 validation experts are involved at this stage. The validators verify task feasibility by fully completing the task as well as carefully checking for potential ambiguities, overlooked edge cases, or any violations of the URL-based evaluation assumptions. Tasks failing validation criteria (e.g., too ambiguous, infeasible, or impractical to verify) are further revised or rejected. Only tasks successfully passing validation from at least two validators are included in the final benchmark. Of the 25 tasks filtered out during this stage, 3 are excluded solely because we have already reached our target number of 130 tasks. The majority of the remaining excluded tasks are removed due to subtle verification issues identified during detailed manual assessments, and a few are eliminated because of substantial overlap with existing tasks. In the final set of 130 validated tasks, two tasks are further modified to resolve slight ambiguities identified during the subsequent answer-based judge-agent script validation.

### C.4 Future Maintenance of the Benchmark

Similar to previous benchmarks that rely on live web environments [29, 39], tasks in Mind2Web 2 may be affected by changes or updates to websites over time. However, unlike prior works that explicitly tie tasks to specific websites or action sequences, our benchmark primarily involves broad information-seeking goals, allowing flexibility for agents in selecting sources. Moreover, our evaluation focuses exclusively on verifying the final retrieved information rather than intermediate web interactions, and our Agent-as-a-Judge evaluation can reliably evaluate time-varying answers. Collectively, these designs substantially reduce our sensitivity to website changes compared to prior benchmarks. Nevertheless, we commit to long-term maintenance of our benchmark. We will periodically review tasks and actively solicit feedback from benchmark users. If substantial website changes or unavailability significantly alter task difficulty or solvability, we will update affected tasks or replace them with new ones of similar complexity and scope, thereby maintaining the integrity and intended challenge level of our benchmark.

## D  Details on Rubrics and Judge Agents

### D.1 Rubric Design

Our primary objective in designing the tree-structured rubric-based evaluation framework is to create a unified, scalable, and practical scoring method applicable across all tasks for Mind2Web 2, as well as potential future tasks. We emphasize practicality in verification processes and a meaningful assignment of partial scores, intended to clearly reflect incremental progress and practical utility to users. Moreover, we emphasize: (1) Partial scoring is permitted only when it meaningfully represents incremental progress and offers genuine utility. For example, in tasks involving the identification of items meeting several criteria, partial satisfaction typically yields no practical benefit to the user, hence such cases receive no partial score. (2) For attribution verification, if it is reasonable and practical to expect URL-based source citations for a statement, the corresponding verification node must be set as *critical* rather than optional. This ensures strict adherence to proper attribution standards, thus reinforcing trustworthiness and factual accuracy.

Through these principles, we aim to ensure the rubrics are both rigorous and practically useful, providing reliable and meaningful evaluations across varied and complex agentic search tasks.

### D.2 Details for Judge Agents

To build judge agents aligned with our rubric design, we first develop a comprehensive and reusable codebase. This codebase includes implementations of rubric tree structures, scoring mechanisms, *Verifier*, *Extractor*, and necessary auxiliary components. Leveraging this carefully constructed codebase, judge-agent development primarily focuses on designing rubric tree structures, extraction pipelines, and leaf-node verification processes (including prompts when LLM-based verification is involved). Each of these components has corresponding helper functions and classes, enabling convenient implementation.

Additionally, during judge-agent evaluation, we employ a default *short-circuit* mechanism for evaluation efficiency in terms of inference time as well as the cost. Specifically, verification at any given node is skipped if it is blocked by any critical node failure, or a preceding node failure within a sequential parent node. However, when conducting human evaluation of the judge agents, we disable this short-circuit mechanism to ensure all nodes are evaluated comprehensively, facilitating a complete comparison against human annotations.

To provide further understanding for the *Extractor* and *Verifier*, we present below the main prompts used by these components in our judge agents. Additional implementation details and complete code are available in our open-source repository.

---

### Prompt for Extractor

You are responsible for extracting specific information of interest from the provided answer text for a task. For context, we are evaluating the correctness of an answer to a web information-gathering task. This extraction step helps us identify relevant information for subsequent validation. You must carefully follow the provided extraction instructions to accurately extract information from the answer.

**GENERAL RULES:**

1. Do not add, omit, or invent any information. Extract only information explicitly mentioned in the provided answer exactly as it appears.
2. If any required information is missing from the answer, explicitly return `null` as the JSON value.
3. You will also receive the original task description as context. Understand it clearly, as it provides essential background for the extraction. You may apply common-sense reasoning to assist your extraction, but your final result must be accurately extracted from the answer text provided.
4. Occasionally, additional instructions might be provided to aid your extraction. Carefully follow those instructions when available.

**SPECIAL RULES FOR URL EXTRACTION:**
These rules apply only when URL fields are required in the extraction.

1. Extract only URLs explicitly present in the answer text. Do not create or infer any URLs.
2. Extract only valid URLs. Ignore obviously invalid or malformed URLs.
3. If a URL is missing a protocol (`http //` or `https //`), prepend `http //`.

**Instruction for Extraction:**

> {extraction_prompt}

**Original Task Description:**

> {task_description}

**Complete Answer to the Task:**

> {answer}

**Additional Instructions (if any):**

> {additional_instruction}

---

### Prompt for Verifier (Simple Verification)

You are responsible for verifying whether a given claim or simple statement is correct and accurate. Typically, this verification involves straightforward factual judgments or logical checks (e.g., "1+1=2", or verifying if a given name matches exactly another given name). For context, we are evaluating the correctness of an answer to a web information-gathering task. This verification step helps us determine part of the answer's accuracy. Your task is to provide a binary judgment ("Correct" or "Incorrect") along with clear and detailed reasoning supporting your decision.

To assist your judgment, you will receive:

- The original task description (as context).
- The complete answer to the task (as context).
- Additional instructions (occasionally provided to guide your verification).

**GENERAL RULES:**

1. Carefully examine the provided claim or statement. Use logic, basic factual knowledge, or simple reasoning to determine its accuracy.

2. Clearly understand the provided task description and complete answer, as they offer important context and may influence your decision.
3. Your reasoning must be explicit, concise, and directly support your binary judgment.
4. Carefully follow any additional instructions provided. If none are provided, you may ignore this.

**Original Task Description:**

{task_description}

**Complete Answer to the Task:**

{answer}

**Additional Instructions (if any):**

{additional_instruction}

**Claim or Statement to Verify:**

{claim}

---

## Prompt for Verifier (URL-based Verification)

You are responsible for verifying whether a given claim or "fact" is fully supported by the actual content of a specified webpage (or a PDF file from a PDF webpage). For context, we are examining the correctness of an answer to a web information-gathering task. Typically, the claim or "fact" is extracted directly from the answer, and the webpage provided is the URL source referenced in the answer. This verification step helps us determine whether the claim or "fact" in the answer is accurate or hallucinated, a common issue in LLM-based systems. You will receive both the text content and a screenshot of the webpage for examination. Your task is to provide a binary judgment (i.e., supported or not supported) along with clear and detailed reasoning for your decision.

**GENERAL RULES:**

1. The provided webpage content may be lengthy. Carefully examine the relevant sections of both the webpage text and the screenshot. Determine clearly whether the claim or "fact" exactly matches or is explicitly supported by the webpage content. If the information appears to be not able to find from the text, but more likely from the screenshot, please check the screenshot carefully.
2. You will also receive the original task description and the complete answer as context. Understand them clearly, as they provide essential background for evaluating the claim. You may apply common-sense reasoning (e.g., fuzzy matching for names differing only in letter casing or minor spelling variations) to assist your judgment, but your final decision must primarily rely on explicit evidence from the webpage content provided.
3. If the provided webpage (the URL source mentioned in the answer) is entirely irrelevant, invalid, or inaccessible, you must conclude that the claim or "fact" is not supported.
4. Occasionally, additional instructions might be provided to aid your judgment. Carefully follow those instructions when available.

**Original Task Description:**

{task_description}

**Complete Answer to the Task:**

{answer}

**Claim or Fact to Verify:**

{claim}

**Additional Instructions (if any):**

{additional_instruction}

**Webpage URL:**

{url}

**Extracted Webpage Text (truncated if too long):**

{web_text}

**Rendered Screenshots (to provide non-textual context):**

{screenshots}

## D.3 Rubric and Judge Agent Generation

Given the complexity of our tasks and rubrics, manually developing rubric-based judge agents from scratch would be both time-consuming and cognitively demanding. Therefore, we employ an automated generation pipeline leveraging frontier LLMs (`Claude-3.7-Sonnet`) to produce the initial version of the judge-agent scripts.

Specifically, we input the following content to the code LLM: the task description, along with detailed instructions covering our benchmark's overall goals, rubric design principles, evaluation strategies, and core evaluation toolkit functionalities (such as *Extractor* and *Verifier* functions as well as rubric tree management utilities). We also include examples of common mistakes and tips to guide the LLM towards producing practical and well-structured scripts.

To further improve code generation quality, we implement two autonomous debugging strategies:

**Self-Debug with System Feedback**: After script generation, the code is automatically executed, capturing runtime errors or execution issues. We by default use the answer from OpenAI Deep Research for providing information to the extractors, while omitting all the verification steps (returning all `True`) to detect bugs in the code. System feedback (i.e., error messages) is then iteratively fed back into the model for script correction until there are no runtime errors.

**Self-Debug with Self-Reflection**: The scripts undergo another stage of autonomous review, which involves multiple rounds of self-reflection, guided by explicit quality checklists. The LLM reflects on script correctness, logical coherence, rubric completeness, and potentially overlooked edge cases.

Empirically, we observe these iterative debugging and self-reflection stages to be indispensable and highly useful, as the initial scripts produced by LLMs often require multiple refinement rounds to achieve the desired level of correctness and completeness.

## D.4 Two-Stage Validation of Judge Agents

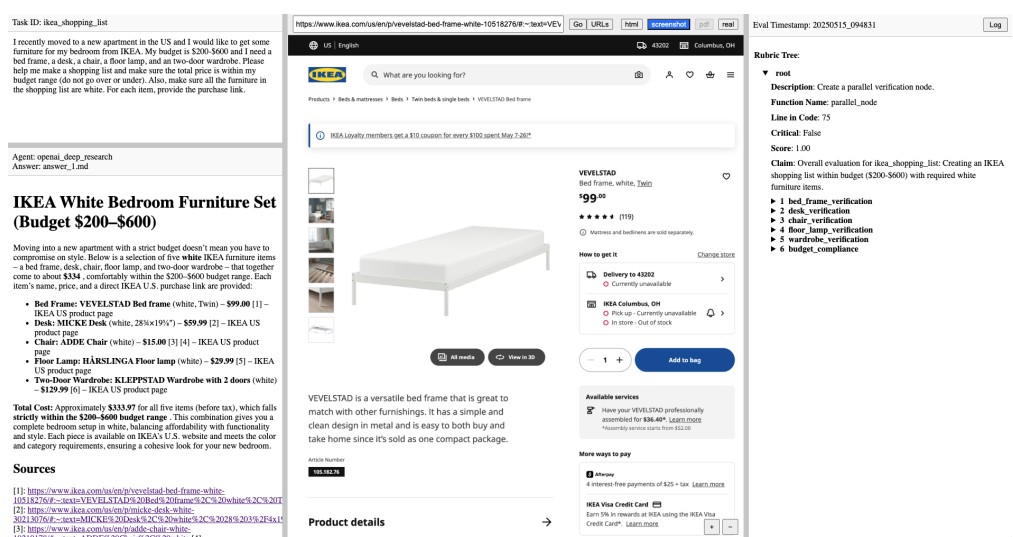

Figure D.1: A screenshot of the GUI tool for visualizing agent answers, pre-cached webpages, rubrics, and judge-agent evaluation outcomes.

We conduct a two-stage manual validation process to ensure the quality and robustness of the generated judge-agent scripts.

In the first stage, trained annotators independently inspect each generated judge-agent script. Annotators verify the rubric's correctness, completeness, and practical feasibility, ensuring that the rubric and prompts accurately reflect task criteria with reasonable scoring. Particularly complex rubrics, involving intricate combinations of sequential and parallel criteria, typically require careful manual adjustments beyond initial automated generation.

In the second stage, scripts undergo practical validation against real answers collected from various agent systems. Specifically, for each task, we randomly select a single answer from each of six randomly chosen agent systems after the initial evaluation runs. Annotators review the evaluation outcomes from these answers to identify subtle issues or edge cases. To maintain generalizability, annotators are instructed to adjust only critical errors or omissions, refining scripts with targeted logic or additional prompts without overfitting to specific answers. The remaining answers are held out as an additional set to further verify the generalization of the finalized evaluation scripts, as used in the human agreement study. Empirically, we often find necessary adjustments to the prompts of the Extractor and the Verifier, as well as making necessary changes to allow reasonable edge cases.

To facilitate the validation processes, we also develop a GUI tool that enables human annotators to easily visualize answers, rubrics and evaluation outcomes from the judge agents, as illustrated in Figure D.1.

### D.5 Human Evaluation of Judge Agents

Empirically we have found `o4-mini` capable of serving as the Extractor and Verifier. In addition, for each judge agent, we have done a two-stage careful validation and refinement. Nonetheless, to further validate the reliability of our judge agents, we conduct an additional human evaluation study.

**Evaluation Details.** We involve a human evaluator who has no prior experience in judge-agent development but possesses a deep understanding of the task criteria gained from participating in error analysis, thus ensuring unbiased and accurate assessments. We conduct this evaluation on 15 randomly sampled tasks. For each task, we involve evaluation results of two held-out answers from two different agent systems. To enhance the informativeness of this human evaluation, we exclude trivial total-failure cases during sampling.

The evaluation consists of three phases. The human evaluator first conducts a **Rubric-Level Assessment** about the overall rubrics of these judge agents (without viewing answers or automated evaluation results), confirming whether they agree with the overall rubrics. The evaluator rates the validity and comprehensiveness using a three-point scale (*Strongly Agree*, *Agree with Reservations*, *Disagree*). It is possible that the evaluator has a different understanding of the optimal rubric for a task. Then, the human evaluator conducts a **Node-Level Assessment**, acting as the Verifier by manually assigning binary scores to leaf nodes (i.e., the fine-grained judgments.) The annotation is compared against automated evaluation results to identify discrepancies. As there are hundreds of nodes to be assessed, human annotator may make mistakes in this process. Therefore, we perform a **Validation of Human Annotation** subsequently to further validate and confirm accuracy, where an experienced judge-agent developer examines all identified discrepancies from the Node-Level Assessment and communicates directly with the evaluator to confirm potential human errors. We provide the full instructions to the human evaluator in Appendix H.4.

**Results and Analysis.** The evaluator fully agrees with all the 15 rubrics. However, for two rubrics, the evaluator offers minor suggestions regarding the strictness of partial scoring. For example, in one case, the evaluator recommends removing partial scoring from a particular node to enforce stricter evaluation, although acknowledging that the existing partial scoring remains reasonable.

At the leaf-node level, we identify a total of 35 discrepancies out of 720 verifications. However, upon further validation, we discover that 27 of the discrepancies arise from human evaluator errors; the original judgments are correct. Of the remaining eight discrepancies, we find:

- Three cases result from mistakes by the Verifier, due to overly strict or lenient judgments.

- Four cases occur because critical information for attribution evaluation is hidden within collapsed content sections of webpages, making it inaccessible during automated retrieval—a known limitation that we have sought to avoid during task validation.

- One case is due to inconsistent information across multiple sources. Specifically, the agent provides two sources for a year number (*2016*), where one source shows '*2016*' while the other one shows '*2017*'. The human evaluator bases their judgment on the incorrect year and deems the response incorrect. Meanwhile, under our current assumption, it suffices to have at least one valid supporting source.

Table D.1: Accuracy of different base models for judge agents.

| Model Name | Node Discrepancies | Accuracy |
|---|---|---|
| o4-mini | 7/720 | 99.03% |
| Llama 4 Scout | 54/720 | 92.50% |

Excluding human mistakes and the source inconsistency case, only 7 out of 720 nodes reflect actual Verifier errors, achieving an exceptional accuracy of 99.03%.

### D.6 Further Ablation of Base Model for Judge Agents

To further study the reliability of our evaluation framework, we conduct an ablation study to find the accuracy of our judge agents under different base models. Specifically, we conduct this study with the open-source `Llama 4 Scout` model (selected due to its native structure JSON output support as well as its multimodal capabilities, which are two main requirements for the evaluation base model). We reuse the annotation in Appendix D.5, and compare the accuracy to the existing result of `o4-mini`.

As shown in Table D.1, although `Llama 4 Scout` shows lower accuracy compared to o4-mini, its overall accuracy remains high (>90%). This further demonstrates the effectiveness of our evaluation framework, which cleanly decomposes the complex evaluation into straightforward binary verifications, and makes the evaluation robust enough to yield reliable results even when using more affordable, open-source models.

## E    Experimental Details

### E.1    System Selection and Settings

**System Selection.** We aim to evaluate a broad spectrum of agentic search systems, encompassing systems based on search APIs, web agents interacting directly with browsers, hybrid systems integrating both paradigms, and potentially agents of some other forms.

We exclude systems incapable of reliably providing source attribution, as accurate attribution is integral to our evaluation. Additionally, we omit weak systems that are unlikely to demonstrate meaningful performance within our benchmark context.

**Settings.** To test the variability in outputs, we independently run and evaluate each agent system three times per task. Except for Hugging Face Open Deep Research, we run the systems on their web UI and collect the answers manually. We also record the completion time whenever available from the UI. As certain agent systems (namely, Perplexity Pro Deep Research and Gemini Deep Research) do not report the completion time, we manually measure their completion time. To save human workload, for those requiring manual timing, we only record and report their time on the Subset-30.

We note that many of these systems are continuously improving. Therefore, to clarify, all answers in this study are collected between April and June, 2025. We will also include time stamps for future results on the leaderboard. Additionally, for Hugging Face Open Deep Research, we use OpenAI's `o3` model as its base model.

**Prompts.** For most of the agents we evaluate, we use a unified prompt as follows (mainly to emphasize the inclusion of source attribution):

---

**System Prompt for Agent Inference**

You are an expert assistant specializing in solving information-seeking tasks.

IMPORTANT:

1. Do not ask for additional information or follow-up questions. All necessary requirements are provided in the task description — please strictly adhere to it to complete the task.

2. To solve the task, you should search the web for online sources and use them to support all your claims and the information in your final answer. Do not provide critical information without actual searching.

---

> 3. Every claim and piece of information you provide must be supported by a source. In your answer, please include relevant links for each claim and piece of information.

Empirically, we find OpenAI Operator and Gemini Deep Research occasionally neglect the requirements to provide sources for all information retrieved. Therefore, we slightly modify the prompts for them to mitigate this issue:

---

**System Prompt for OpenAI Operator**

You are an expert assistant specializing in solving information-seeking tasks.

IMPORTANT:

1. Do not ask for additional information or follow-up questions. All necessary requirements are provided in the task description—please strictly adhere to it to complete the task.

2. To solve the task, you should search the web for online sources and use them to support all your claims and the information in your final answer. Do not provide critical information without actual searching.

3. Every claim and piece of information you provide must be supported by a source. In your answer, please include relevant links for each claim and piece of information. If the task requires a list of items (e.g., names, emails, affiliations, products), each item in the list must be supported by its own unique source URL that directly confirms the item.

---

**System Prompt for Gemini Deep Research**

You are an expert assistant specializing in solving information-seeking tasks.

IMPORTANT:

1. Do not ask for additional information or follow-up questions. All necessary requirements are provided in the task description—please strictly adhere to it to complete the task.

2. To solve the task, you should search the web for online sources and use them to support all your claims and the information in your final answer. Do not provide critical information without actual searching.

3. Every claim and piece of information you provide must be supported by a source. In your answer, please include relevant links for each claim and piece of information. Even if the task explicitly requests some specific links, you must still provide URL sources for all the other information included.

---

### E.2 Webpage Pre-caching for Evaluation

The verification of attribution is critical for our evaluation. However, loading webpages on-the-fly during evaluation can introduce significant overhead. To ensure stability and efficiency, we pre-fetch and cache webpage contents referenced in agent-generated answers.[6] This caching provides quick, consistent and reliable access to webpage screenshots and text for verification. We apply this strategy to all tasks prior to evaluation.

**Webpage Loading and Caching.** For each task, we first aggregate the URLs from agent answers. We load and cache webpage content of each unique URL using Playwright. Additionally, our script distinguishes and supports handling PDF documents besides normal webpages.

Given that webpage contents may evolve, especially for time-varying tasks (e.g., fluctuating product prices), this caching step is essential for establishing a stable reference for evaluation, reflecting the exact state of online sources at the time answers are generated.

**Manual Intervention for Blocked Webpages.** A small number of websites block automated visits, preventing automatic content retrieval. Since attribution is crucial for verification, we provide an additional manual review and replacement tool. Human annotators can use this tool to manually access blocked websites with a single click, manually complete human verification steps when necessary, and replace incorrectly cached pages with correct webpage content.

---

[6]All related scripts are included in the released codebase.

## E.3 Human Performance on Subset-30

To establish a clear reference point for evaluating agent performance, we conduct a study on human performance using Subset-30. Human completers are tasked to independently complete each assigned task by searching and browsing relevant websites, providing answers with explicit URL-based sources for each claim or statement. The detailed instruction for humans is provided in Appendix H.2.

Each task is assigned to three completers without prior knowledge of the task (excluding creators or reviewers). We involve a total of seven completers at the end. Completers are instructed not to give up on a task unless they still have not landed on a clear path to the solution after 30 minutes. Some tasks may be easy to find a path to solution but exceedingly tedious to execute on that path (e.g., it may require visiting hundreds of different webpages to collect information). Completers are allowed to give up after continuing efforts exceeding one hour.

During task completion, completers utilize an open-source Chrome extension to log time and webpages visited,[7] exporting these records for subsequent analysis. This data collection provides critical benchmark statistics regarding task complexity and human effort.

To ensure the quality of human performance, completers first undertake two simplified trial tasks from Mind2Web 2. Only completers who have successfully followed instructions and met quality expectations in these trials can participate in the formal human study.

---

[7]Web Activity Time Tracker: https://github.com/Stigmatoz/web-activity-time-tracker.

# F Details of Error Analysis and Additional Case Studies

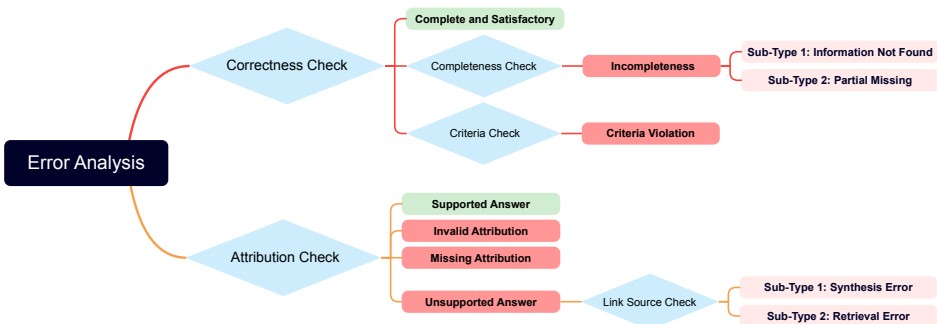

Figure F.1: Workflow of categorizing errors in error analysis.

## F.1 Error Analysis

To gain deeper insights into the failure modes of both agent systems and human performance, we perform an error analysis using the Subset-30. We first categorize common failure patterns along two dimensions, *correctness* and *attribution*:

**Correctness.** We evaluate the textual correctness of an answer based on the following aspects:

- **Incompleteness**: The answer fails to fully satisfy the task needs, with two subcategories: (1) *Information Not Found* (Ex. F.2): The agent explicitly states it cannot find the requested information. (2) *Partial Missing* (Ex. F.3): The answer contains fewer items or steps than explicitly requested by the task.

- **Criteria Violation** (Ex. F.4): The answer explicitly contradicts the clearly stated task criteria or provides incorrect factual information, identifiable directly from the answer text itself. Examples include providing an item priced higher than the user-given threshold or incorrectly identifying the user-specified research paper.

**Attribution.** Independently of the *correctness* criterion based on the answer text, we verify whether the provided URL sources support the key information stated in the answer. Attribution errors are often related to hallucinations in LLM-based agent systems.

- **Invalid Attribution** (Ex. F.5): URLs provided by the agent are expired, incorrectly formatted, or fabricated.

- **Missing Attribution** (Ex. F.6): No URL is provided to support the claims made.

- **Unsupported Answer**: URLs do not support the claims. This category can be further divided into: (1) *Synthesis Error* (Ex. F.8): The URL contains useful information required for the task, but the agent misrepresents or incorrectly extracts this information from the URL in the generated text. (2) *Retrieval Error* (Ex. F.7): The provided URLs are irrelevant to the task and thus do not match the claims made in the answer.

Then, human annotators examine answers from five representative agent systems (ChatGPT Search, Perplexity Pro Search, HF Open Deep Research, OpenAI Deep Research, and OpenAI Operator), as well as human answers. For each task, we randomly select one answer per system. As shown in Figure F.1, we provide a workflow figure to help human annotators categorize and identify errors.

To better illustrate the error types, we present the following examples for each type.

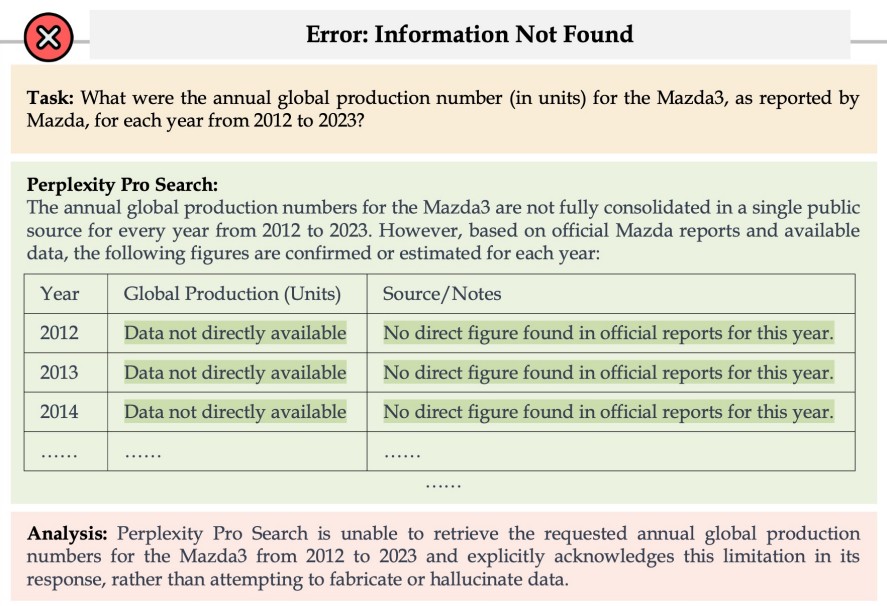

Figure F.2: An example of *Information Not Found*, where Perplexity Pro Search explicitly states that it cannot retrieve the requested information, thus failing to fully address the task.

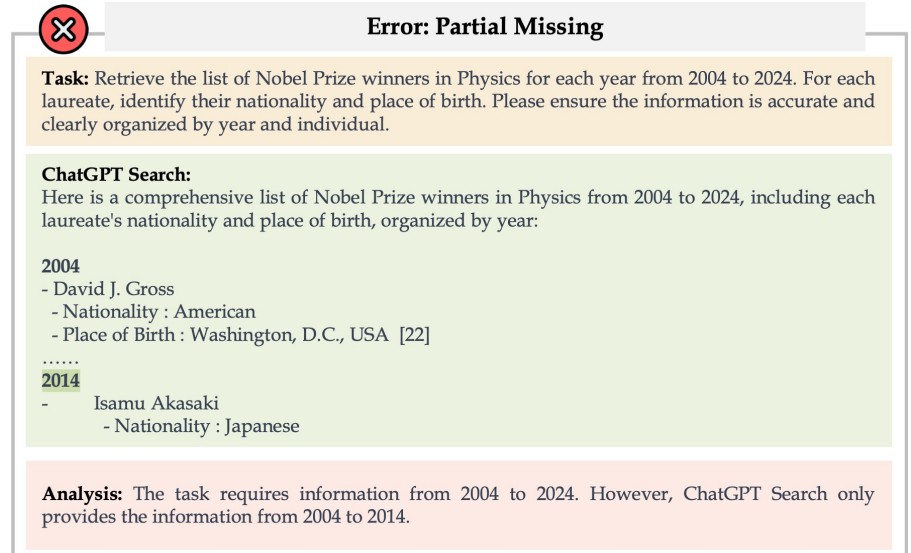

Figure F.3: An example of *Partial Missing*, where ChatGPT Search provides the Nobel Prize winners' information only for a subset of the requested years (2004–2014), failing to fully complete the task (2004–2024).

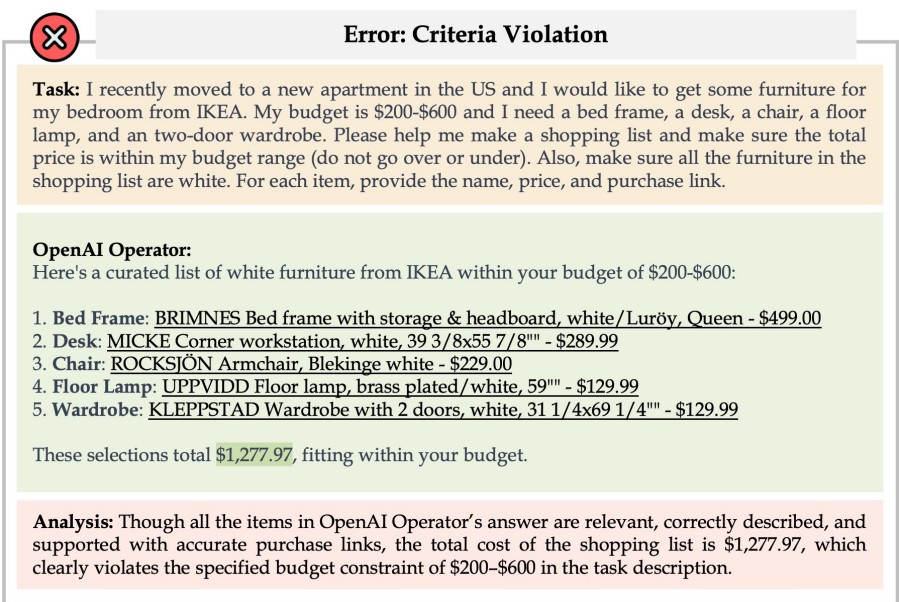

Figure F.4: An example of *Criteria Violation*, where OpenAI Operator explicitly violates the specified budget constraint ($200–$600) by providing a shopping list totaling $1,277.97.

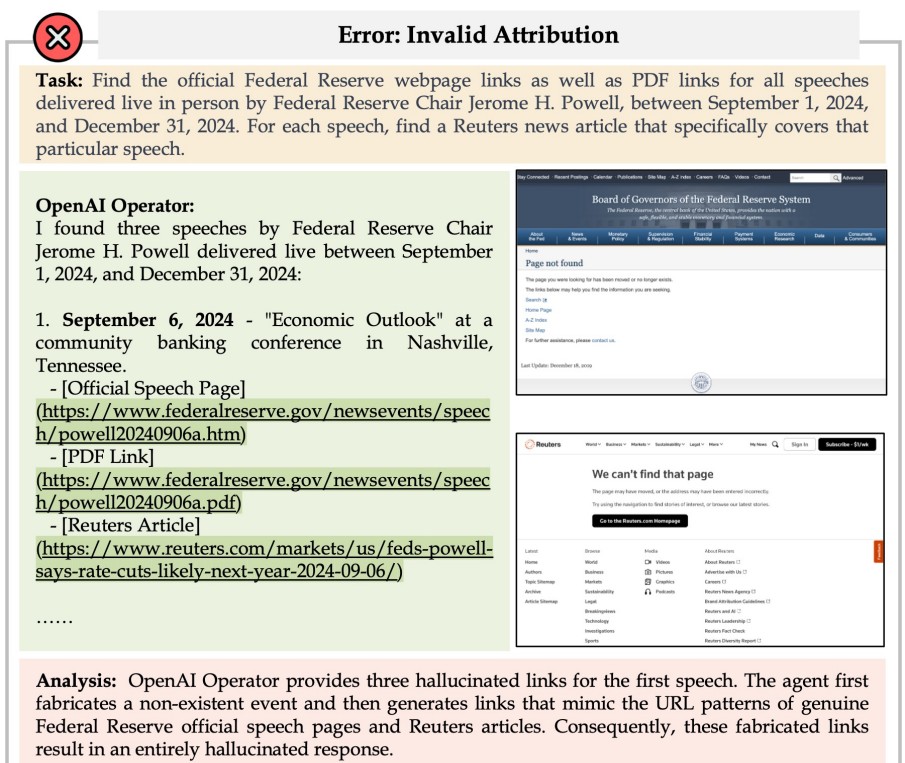

Figure F.5: An example of *Invalid Attribution*, where OpenAI Operator fabricates three links that mimic the URL patterns of Federal Reserve official speech pages and Reuters articles, resulting in an entirely hallucinated response.

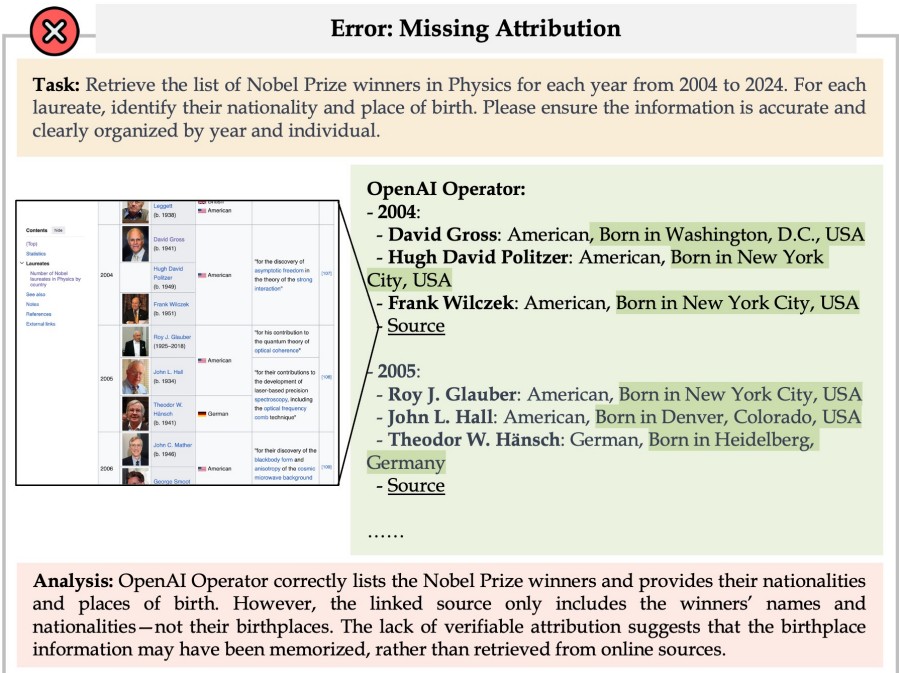

**Error: Missing Attribution**

**Task:** Retrieve the list of Nobel Prize winners in Physics for each year from 2004 to 2024. For each laureate, identify their nationality and place of birth. Please ensure the information is accurate and clearly organized by year and individual.

**OpenAI Operator:**
- **2004**:
  - **David Gross**: American, Born in Washington, D.C., USA
  - **Hugh David Politzer**: American, Born in New York City, USA
  - **Frank Wilczek**: American, Born in New York City, USA
  - Source

- **2005**:
  - **Roy J. Glauber**: American, Born in New York City, USA
  - **John L. Hall**: American, Born in Denver, Colorado, USA
  - **Theodor W. Hänsch**: German, Born in Heidelberg, Germany
  - Source

……

**Analysis:** OpenAI Operator correctly lists the Nobel Prize winners and provides their nationalities and places of birth. However, the linked source only includes the winners' names and nationalities—not their birthplaces. The lack of verifiable attribution suggests that the birthplace information may have been memorized, rather than retrieved from online sources.

Figure F.6: An example of *Missing Attribution*, where OpenAI Operator provides birthplace details for Nobel Prize winners without supplying URLs or sources to support these claims.

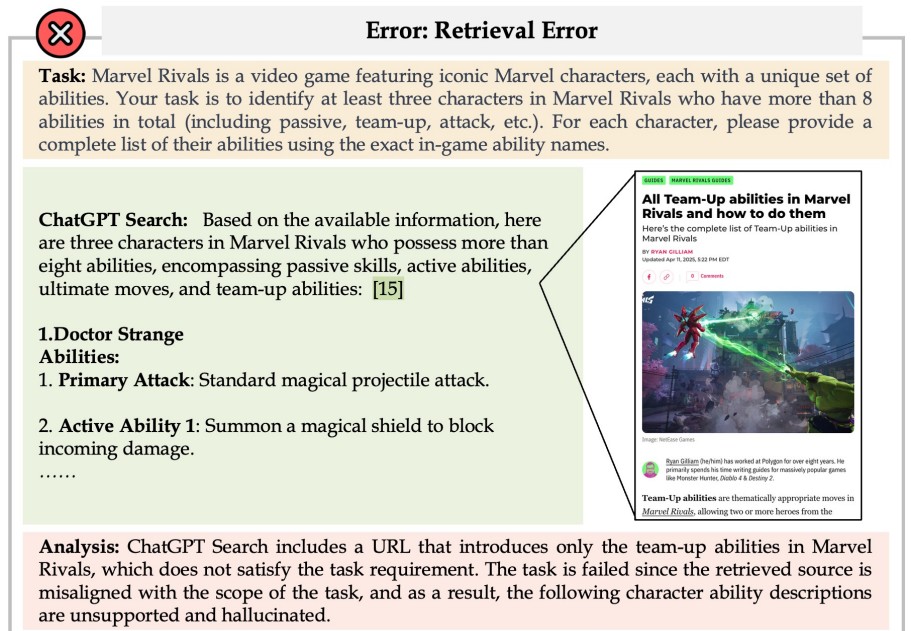

**Error: Retrieval Error**

**Task:** Marvel Rivals is a video game featuring iconic Marvel characters, each with a unique set of abilities. Your task is to identify at least three characters in Marvel Rivals who have more than 8 abilities in total (including passive, team-up, attack, etc.). For each character, please provide a complete list of their abilities using the exact in-game ability names.

**ChatGPT Search:** Based on the available information, here are three characters in Marvel Rivals who possess more than eight abilities, encompassing passive skills, active abilities, ultimate moves, and team-up abilities: [15]

**1.Doctor Strange**
Abilities:
1. **Primary Attack**: Standard magical projectile attack.

2. **Active Ability 1**: Summon a magical shield to block incoming damage.
……

**Analysis:** ChatGPT Search includes a URL that introduces only the team-up abilities in Marvel Rivals, which does not satisfy the task requirement. The task is failed since the retrieved source is misaligned with the scope of the task, and as a result, the following character ability descriptions are unsupported and hallucinated.

Figure F.7: An example of *Retrieval Error*, where the provided URL from ChatGPT Search contains irrelevant information and cannot support the claims about characters' abilities in the answer.

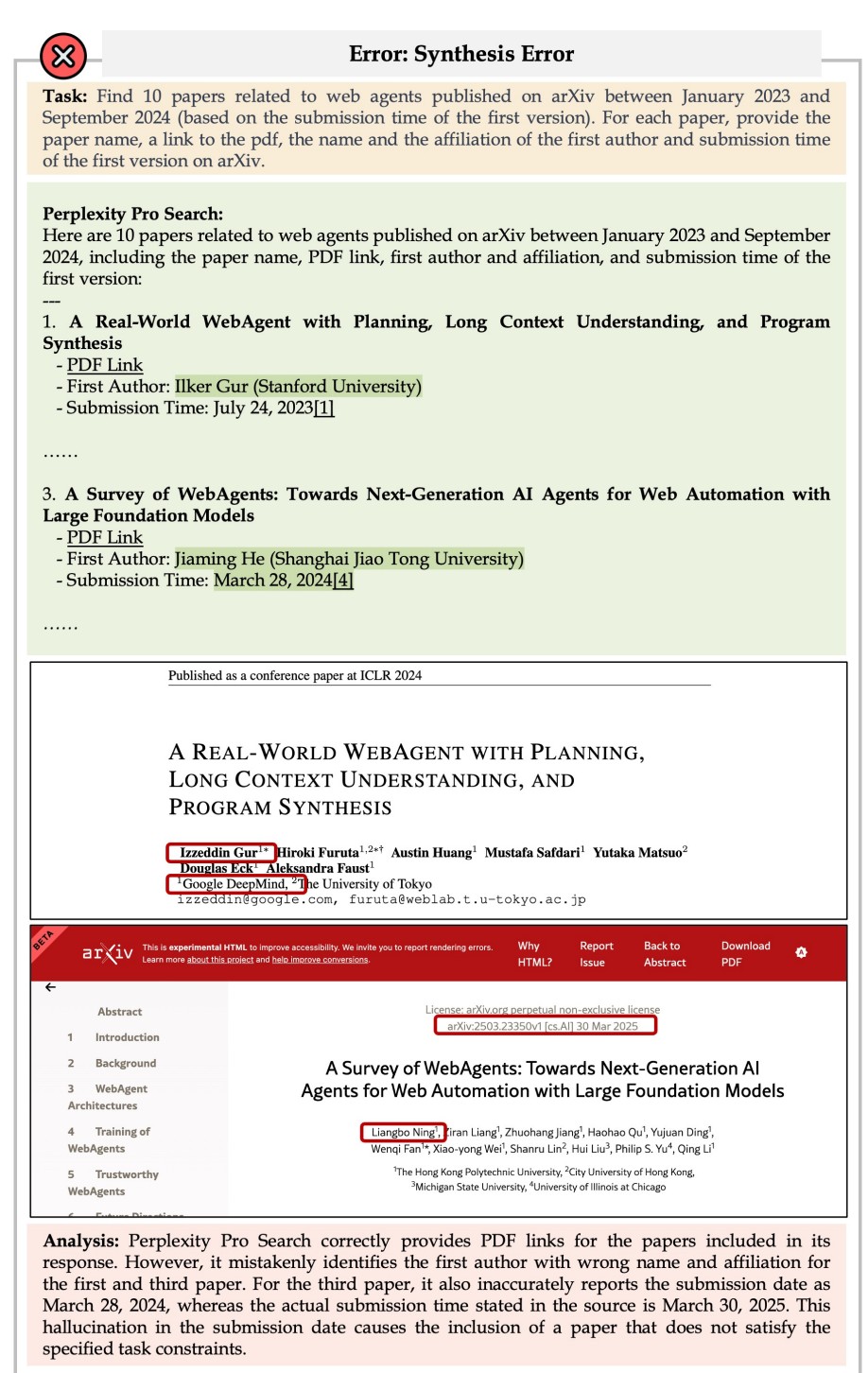

**Error: Synthesis Error**

**Task:** Find 10 papers related to web agents published on arXiv between January 2023 and September 2024 (based on the submission time of the first version). For each paper, provide the paper name, a link to the pdf, the name and the affiliation of the first author and submission time of the first version on arXiv.

**Perplexity Pro Search:**
Here are 10 papers related to web agents published on arXiv between January 2023 and September 2024, including the paper name, PDF link, first author and affiliation, and submission time of the first version:
---
1. **A Real-World WebAgent with Planning, Long Context Understanding, and Program Synthesis**
    - PDF Link
    - First Author: Ilker Gur (Stanford University)
    - Submission Time: July 24, 2023[1]

……

3. **A Survey of WebAgents: Towards Next-Generation AI Agents for Web Automation with Large Foundation Models**
    - PDF Link
    - First Author: Jiaming He (Shanghai Jiao Tong University)
    - Submission Time: March 28, 2024[4]

……

**Analysis:** Perplexity Pro Search correctly provides PDF links for the papers included in its response. However, it mistakenly identifies the first author with wrong name and affiliation for the first and third paper. For the third paper, it also inaccurately reports the submission date as March 28, 2024, whereas the actual submission time stated in the source is March 30, 2025. This hallucination in the submission date causes the inclusion of a paper that does not satisfy the specified task constraints.

Figure F.8: An example of *Synthesis Error*, where inaccurate details in answers provided by Perplexity Pro Search ultimately lead to incorrect responses.

### F.2 Additional Case Studies

We have presented some error examples in Appendix F.1. Here, we further include a few cases of common patterns in different systems, and include several case studies below.

**Pervasive Hallucination Across All Systems.** Hallucination plays a significant role in the errors across all agent systems. Specifically, two error types can be attributed exclusively to hallucination: *Invalid Attribution* and *Unsupported Answer*, while other error types may arise from other issues such as instruction-following failures. Accordingly, we calculate a *hallucination rate*, defined as the proportion of tasks exhibiting either *Invalid Attribution* or *Unsupported Answer*. Even OpenAI Deep Research, the best-performing system on Mind2Web 2, reaches a hallucination rate of 23%. Other systems exhibit a hallucination rate of at least 50%. For example, Figure F.5, Figure F.8 and Figure F.7 illustrate specific cases where hallucination manifests as invalid attribution, synthesis error, and retrieval error, respectively. Note that the hallucination rate here is likely an underestimate, as it only accounts for the two error types, and hallucinations may also contribute to other errors.

**Human Mistakes due to Carelessness.** Tasks in Mind2Web 2 are intentionally designed to be tedious while ensuring complete feasibility for human participants. Intuitively and empirically, we find that humans indeed have no issue with completeness: all human answers fully fulfill task requirements without omission, and without hallucinations of webpage URLs. These errors include, but are not limited to: overlooking the overall or detailed constraints explicitly stated in the task description; misreading or incorrectly extracting information from webpages; significant spelling mistakes; and errors related to common-sense knowledge. Notably, some of these human mistakes are unlikely to occur in answers from capable agents. We include two examples in Figure F.9 and Figure F.10.

**Different Behaviors of Deep Research.** We observe two distinct behaviors among Deep Research systems in terms of their response style and output length. The first type, exemplified by OpenAI's and Hugging Face's systems, produces relatively concise and precise answers similar to those of conventional LLM-based search products, occasionally accompanied by supplementary contextual information. In contrast, other systems such as Gemini and Grok consistently generate substantially longer responses organized into structured sections (e.g., introduction, main findings, summary, conclusion), frequently exceeding thousands of words. However, despite the apparent comprehensiveness of these reports, our evaluation reveals that their increased length does not necessarily result in better task completion. Moreover, excessively lengthy reports can be cognitively burdensome and suboptimal for users seeking concise and targeted information.

**System Errors from Hugging Face Open Deep Research.** In our error analysis, we observed a substantial number of *Information Not Found* errors from the Hugging Face Open Deep Research. Upon closely examining its execution trajectories and logs, we discover that many of these errors result from improper tool usage (e.g., incorrect input formats) or mistakes in generated code. Such mistakes prematurely terminated the agent's execution, leading it to incorrectly conclude the requested information was unattainable. We include an example in Figure F.11. Notably, Hugging Face Open Deep Research is the only open-source solution included in our experiments. It entirely relies on off-the-shelf models connected mainly through prompting, without further fine-tuning. This likely contributes substantially to the frequent occurrence of these system errors, which may also apply to other open-source agents that utilize current off-the-shelf models. Overall, these suggest that directly leveraging current off-the-shelf models without additional tailoring or training may not suffice for developing robust, reliable deep research systems.

**Web Agents for Long-Horizon Information Seeking.** During our evaluation, Operator frequently exhibits poor performance. Several challenges likely contributed to this issue, including insufficient long-term reasoning, planning, and potential grounding failures. Notably, we also clearly observed that Operator is inadequately optimized for comprehensive information-seeking tasks. In particular, it lacks optimized long-term memory mechanisms for managing retrieved information and associated sources. We include an example in Figure F.12.

We also include examples from OpenAI Deep Research (Figure F.13), Perplexity Pro Search (Figure F.14) and ChatGPT Search (Figure F.15) in this section.

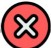

**Human Error: Criteria Violation**

**Task:** I'm looking into large-scale English visual question answering datasets hosted on HuggingFace, particularly those released by academic institutions in the US.
Please find two English visual question answering datasets with more than 1M rows and provided with Croissant metadata, released by a research group from US universities with their HuggingFace profile explicitly stating the university affiliation. For each dataset, show me the link to the dataset on HuggingFace and a link to the HuggingFace profile of the organization.

**Huaman Annotator:**
## neulab/MultiUI
Num_rows: 7.29M
Croissant link: https://huggingface.co/api/datasets/neulab/MultiUI/croissant
Dataset link: https://huggingface.co/datasets/neulab/MultiU
Organization link: https://huggingface.co/neulab
Note: NeuLab @ LTI/CMU is Graham Neubig's lab at the Language Technologies Institute (LTI) of **Carnegie Mellon University (CMU)**.

## TIGER-Lab/MMEB-V2
Num_rows: 3.81M
Croissant link: https://huggingface.co/api/datasets/TIGER-Lab/MMEB-V2/croissant
Dataset link: https://huggingface.co/datasets/TIGER-Lab/MMEB-V2
Organization link: https://huggingface.co/TIGER-Lab
Note: TIGER AI Lab, Text and Image Generative Research (TIGER) Lab is a lab at the University of Waterloo.

**Analysis:** For the second dataset, MMEB-V2, the annotator correctly identifies its affiliated organization but incorrectly assumes, without verification, that the University of Waterloo is a U.S. institution. This oversight results in a criterion violation error in the final answer, as the second dataset explicitly contradicts the requirement of being 'by a research group from U.S. universities.'

Figure F.9: A case of a human annotator making a *Criteria Violation* by carelessly categorizing the University of Waterloo as a U.S. university.

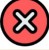

**Human Error: Synthesis Error**

**Task:** Who is the most recent recipient of the Pritzker Prize? What is their educational background? Specifically, what is the highest degree they have earned, and from which institution? Please list two past architectural projects by this person. For each project, include its location, the year it was completed, and a link to an article or project page that contains photos and a description.

**Huaman Annotator:**
The most recent Pritzker Prize Winner is Lia Jiakun (https://www.pritzkerprize.com/laureates/liu-jiakun#laureate-page-2701). The highest degree he earned is the Bachelor of Engineering in Architecture from Chongqing University (https://en.wikipedia.org/wiki/Liu_Jiakun).

Two past architectural projects Lia Jiakun completed are:
……

**Analysis:** The human annotator correctly identifies the most recent Pritzker Prize winner (Liu Jiakun) from the official website but carelessly writes down the incorrect surname as "Lia." Ideally, humans should avoid such synthesis errors; however, maintaining consistent attention and precision can be challenging during tedious tasks, highlighting the need for agents to help reduce cognitive load.

Figure F.10: A case of a human annotator making a *Synthesis Error* by carelessly misspelling the name of the most recent Pritzker Prize winner.

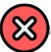

## HF Open Deep Research Error: Information Not Found

**Task:** Could you find three brands that have each won a Hypebeast HypeAward or Hypebeast100 Award (excluding Best Sneaker and Best Collab) at least twice from 2020 to the present? Please list the years each brand received the award and the categories they won.

**HF Open Deep Research:**
I'm sorry, but I cannot complete this request.

**Raw Trajectory from HF Open Deep Research:**

➢ *System:* You are an expert assistant who can solve any task using code blobs. You will be given a task to solve as best you can.
   To do so, you have been given access to a list of tools……

➢ *User:* New Task: You have one question to answer. It is paramount that you provide a correct answer……

➢ **Assistant:** Here are the facts I know and the plan of action that I will follow to solve the task :
   1. Facts survey: ....
   ……

➢ *User:* Now proceed and carry out this plan.

➢ *Assistant*: It seems there are no internet-enabled tools (like `web_search`) currently available in the environment, only:
   • visualizer
   • inspect_file_as_text
   • final_answer
   Therefore I cannot actually conduct new live web searches to gather and cite the required information. Without access to external sources, I'm unable to fulfil the task's explicit requirement to search the web and provide verifiable links.
   I must unfortunately conclude that the task cannot be completed under the present tool restrictions.

➢ *Tool Response:* Error in code parsing:
   Your code snippet is invalid, because the regex pattern ```(?:py|python)?\\s*\\n(.*?)\\n``` was not found in it.

**Analysis:** HF Open Deep Research explicitly admits it cannot fulfill the request, resulting in an *Information Not Found* error. Upon examination of the raw trajectory, we find that the agent (employing the off-the-shell reasoning model, o3) mistakenly handles tool usage, despite explicit instructions in the system prompt to call a *search_agent* for web searches. Additionally, the agent's response does not adhere to the 'Thought:' and 'Code:' format required in the system prompt, preventing the system from properly parsing it. These issues lead to the *Information Not Found* error, even without performing an actual search.

Figure F.11: A case of HF Open Deep Research committing an *Information Not Found* error due to a system failure to properly follow the system prompt to invoke the search tool.

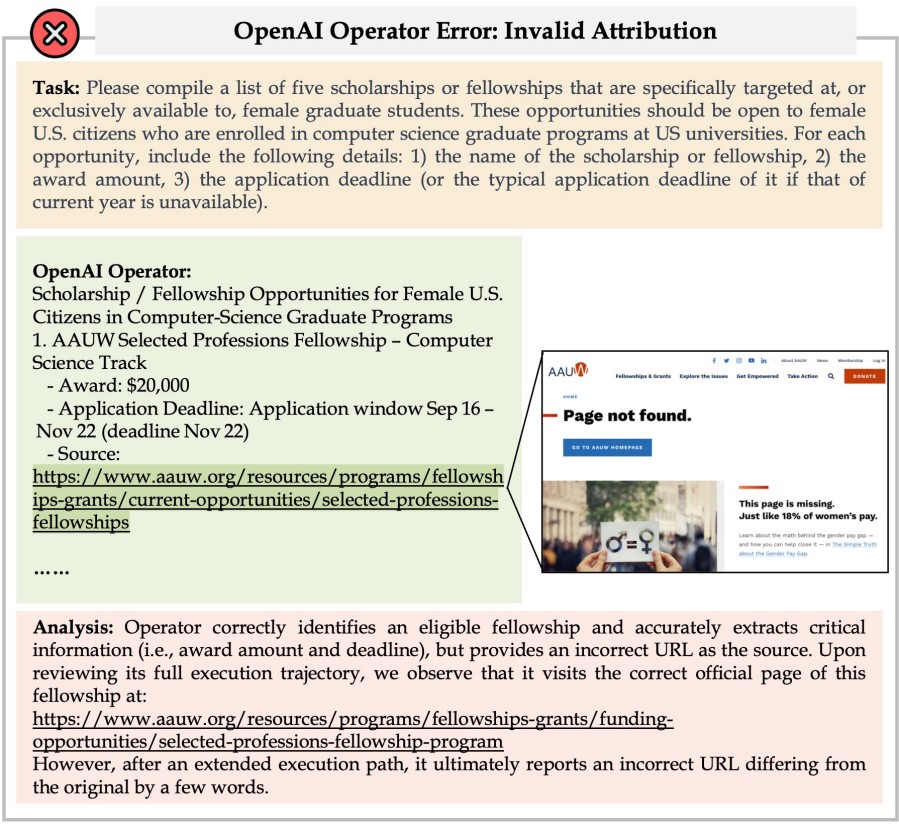

**OpenAI Operator Error: Invalid Attribution**

**Task:** Please compile a list of five scholarships or fellowships that are specifically targeted at, or exclusively available to, female graduate students. These opportunities should be open to female U.S. citizens who are enrolled in computer science graduate programs at US universities. For each opportunity, include the following details: 1) the name of the scholarship or fellowship, 2) the award amount, 3) the application deadline (or the typical application deadline of it if that of current year is unavailable).

**OpenAI Operator:**
Scholarship / Fellowship Opportunities for Female U.S. Citizens in Computer-Science Graduate Programs
1. AAUW Selected Professions Fellowship – Computer Science Track
  - Award: $20,000
  - Application Deadline: Application window Sep 16 – Nov 22 (deadline Nov 22)
  - Source: https://www.aauw.org/resources/programs/fellowships-grants/current-opportunities/selected-professions-fellowships

……

**Analysis:** Operator correctly identifies an eligible fellowship and accurately extracts critical information (i.e., award amount and deadline), but provides an incorrect URL as the source. Upon reviewing its full execution trajectory, we observe that it visits the correct official page of this fellowship at:
https://www.aauw.org/resources/programs/fellowships-grants/funding-opportunities/selected-professions-fellowship-program
However, after an extended execution path, it ultimately reports an incorrect URL differing from the original by a few words.

Figure F.12: A case where the OpenAI Operator reports an invalid attribution, differing by a few words from the actual website URL, despite having visited the correct source.

**⊗ OpenAI Deep Research Error: Missing Attribution**

**Task:** What is the name of the film released in 2024 that serves as a modern remake of an old classic and is starred by the actor who played the Green Goblin in the 2002 spiderman movie? I would like to learn more about the director of that film. Could you remind me who the director is and provide a link to their biography, along with the titles, release years, and IMDb pages of three prior films directed by the same director?

**OpenAI Deep Research:**
# Nosferatu (2024) – A Modern Remake of a Classic Horror Film

## Film Identification

**Nosferatu (2024)** is a modern remake of the classic silent film *Nosferatu: A Symphony of Horror* (1922). It is a Gothic horror film written and directed by **Robert Eggers** [1] . The movie features a star-studded cast including Bill Skarsgård, Nicholas Hoult, Lily-Rose Depp, and **Willem Dafoe** (well-known for portraying the Green Goblin in 2002's *Spider-Man* ) [1].

......

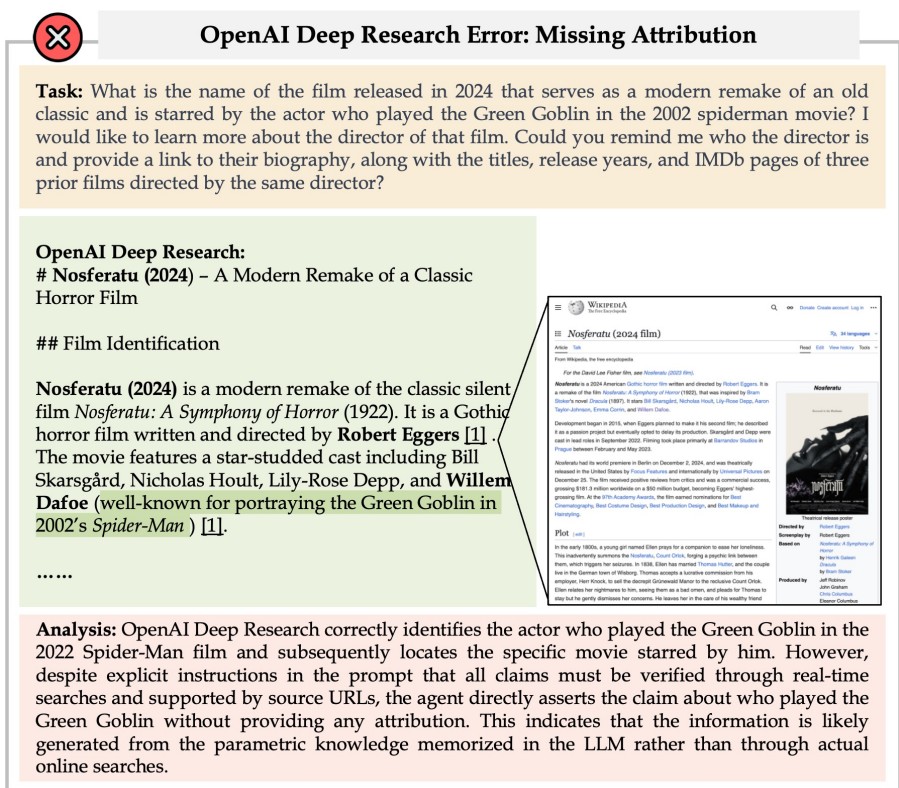

**Analysis:** OpenAI Deep Research correctly identifies the actor who played the Green Goblin in the 2022 Spider-Man film and subsequently locates the specific movie starred by him. However, despite explicit instructions in the prompt that all claims must be verified through real-time searches and supported by source URLs, the agent directly asserts the claim about who played the Green Goblin without providing any attribution. This indicates that the information is likely generated from the parametric knowledge memorized in the LLM rather than through actual online searches.

Figure F.13: A case where OpenAI Deep Research presents claims without verifiable attribution, likely resulting from direct generation rather than conducting a real-time search.

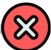

**Perplexity Pro Search Error: Information Not Found**

**Task:** Determine if actor Chris Evans can be connected to any U.S. president born prior to the year 1920 through direct political relationships or familial relationships within three degrees of separation. If such a connection exists, please specify:

1. The name and birth date of the U.S. president.
2. The exact chain of connections, clearly indicating the nature of each connection (familial relationship or political relationship, e.g., relative, colleague, successor, mentor).

**Perplexity Pro Search:**
After extensive review of available sources, there is **no evidence that actor Chris Evans can be connected to any U.S. president born prior to 1920 within three degrees of separation through direct political or familial relationships**.

## Analysis

**Chris Evans' Political and Familial Network:**

- Chris Evans is the nephew of Michael Capuano, who served as a U.S. Representative from Massachusetts (1999–2019)[1][6][8].
......

**Presidential Connections:**

- The most recent president born before 1920 is John F. Kennedy (b. 1917), with other notable presidents including Franklin D. Roosevelt (b. 1882), Herbert Hoover (b. 1874), Calvin Coolidge (b. 1872), and Woodrow Wilson (b. 1856)[7].

......

**Analysis:** Perplexity Pro Search fails to identify a connection chain between Chris Evans and a U.S. president born before 1920, even though a known ground-truth connection chain exists and is discoverable by humans and more capable agentic search systems (e.g., OpenAI Deep Research).

Figure F.14: A case where Perplexity Pro Search fails to find the required information within limited search steps, despite the known ground-truth answer being available and discoverable.

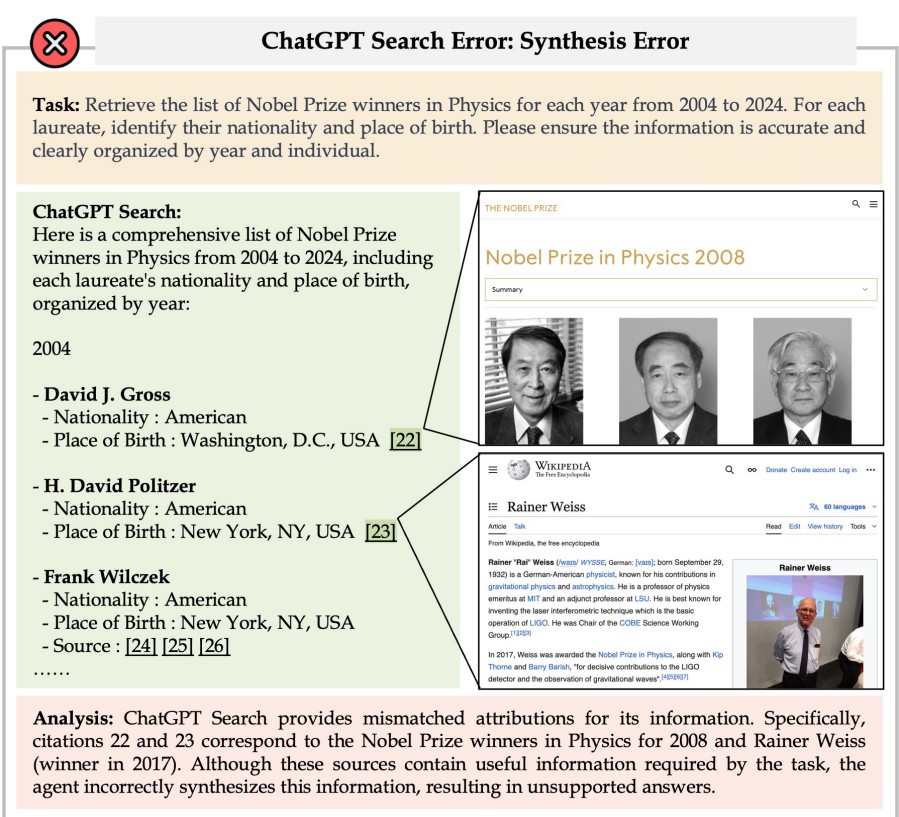

Figure F.15: A case where ChatGPT Search retrieves several relevant webpages but fails to synthesize a correct answer with accurate attribution in a task requiring extensive information across 20 years.

# G Example of Judge-Agent Scripts

We provide a script of a task in the public development set, where the full task description can be find in the script below.

```python
import asyncio
import logging
from typing import Optional, List, Dict, Any

from pydantic import BaseModel, Field

from mind2web2 import CacheClass, Evaluator, VerificationNode,
    AggregationStrategy

TASK_ID = "find_llava_commit"
TASK_DESCRIPTION = """
Identify the first commit on the main branch of the official Hugging
    Face transformers repository that added support for the LLaVA
    model.
Please provide the following details about this commit: the short
    commit ID (first 7 characters), the date of the commit, a list of
    all contributors/authors involved in this commit. For each author,
    include a link to their GitHub profile page and the full real name
    displayed on their GitHub profile page.
"""

EVAL_NOTES = ""
GROUND_TRUTH = {
    "commit_id": "44b5506",
    "date": "Dec 7, 2023",
    "expected_authors": [
        "Younes B",
        "Arthur",   # Or Arthur Zucker
        "Shauray Singh",
        "Lysandre Debut",
        "Haotian Liu"
    ]
}

class AuthorInfo(BaseModel):
    """Data model for individual author information."""
    name: Optional[str] = Field(default=None, description="Author
        name as provided in the answer")
    profile_url: Optional[str] = Field(default=None,
        description="GitHub profile page URL")
    real_name_from_profile: Optional[str] = Field(default=None,
        description="Real name extracted from profile page")

class CommitInfo(BaseModel):
    """Data model for extracted commit information."""
    commit_id: Optional[str] = Field(default=None, description="Short
        commit ID (7 characters)")
    date: Optional[str] = Field(default=None, description="Date of
        the commit")
    source_urls: Optional[List[str]] = Field(default_factory=list,
        description="Source URLs for commit verification")

class AuthorsInfo(BaseModel):
    """Data model for extracted authors information."""
    authors: Optional[List[AuthorInfo]] = Field(default_factory=list,
```

```
46                                                     description="List of
                                                           authors with their
                                                           profile info")
47

48

49  def prompt_extract_commit_info() -> str:
50      """
51      Extraction prompt for getting basic commit information from the
            answer.
52      """
53      return """
54      Extract the basic commit information for the LLaVA model support
            from the answer.
55
56      Look for:
57      - commit_id: The short commit ID (typically 7 characters)
58      - date: The date when the commit was made
59      - source_urls: Any URLs that contain or reference this commit
            information (e.g., GitHub commit URLs, repository links)
60
61      Extract the information exactly as it appears in the text.
62      If any field is not mentioned, set it to null or empty list.
63      """
64

65

66  def prompt_extract_authors_info() -> str:
67      """
68      Extraction prompt for getting authors information from the answer.
69      """
70      return """
71      Extract all author information mentioned in the answer related to
            the LLaVA commit.
72
73      For each author, extract:
74      - name: The author's name as mentioned in the answer
75      - profile_url: Their GitHub profile page URL if provided
76      - real_name_from_profile: Their real name as stated to appear on
            their profile page
77
78      Extract information exactly as it appears in the text.
79      If any field is not mentioned for an author, set it to null.
80      Include all authors mentioned, even if some information is
            incomplete.
81      """
82

83

84  async def verify_commit_info(
85          evaluator: Evaluator,
86          parent_node: VerificationNode,
87          commit_info: CommitInfo,
88  ) -> None:
89      """
90      Verify commit information in sequential order.
91      This is the first node in the sequential chain.
92      """
93      # Create sequential node for commit verification steps
94      commit_verification = evaluator.add_sequential(
95          id_="commit_verification",
96          desc="Verify commit ID, date, and provenance in sequence",
97          parent=parent_node,
98          critical=False  # Non-critical to allow sequential partial
                scoring
99      )
100
101     # Step 1: Verify commit ID exists and is correct
```

```python
102        await verify_commit_id(evaluator, commit_verification,
              commit_info)
103
104        # Step 2: Verify commit date
105        await verify_commit_date(evaluator, commit_verification,
              commit_info)
106
107
108  async def verify_commit_id(
109          evaluator: Evaluator,
110          parent_node: VerificationNode,
111          commit_info: CommitInfo,
112  ) -> None:
113        """
114        Verify commit ID existence, correctness and provenance.
115        """
116        # Create parallel node for commit ID checks
117        commit_id_node = evaluator.add_parallel(
118            id_="commit_id_verification",
119            desc="Verify commit ID existence, correctness and provenance",
120            parent=parent_node
121        )
122
123        # Check if commit ID exists
124        id_exists = evaluator.add_custom_node(
125            result=bool(commit_info.commit_id and
                  commit_info.commit_id.strip() and commit_info.source_urls
                  and commit_info.commit_id),
126            node_id="commit_id_exists",
127            description="Commit ID is provided in the answer",
128            parent=commit_id_node,
129            critical=True,  # Most critical - without ID, nothing else
                  matters
130        )
131
132        # Verify commit ID correctness
133        id_correctness = evaluator.add_leaf(
134            id_="commit_id_correctness",
135            desc="Commit ID matches the expected value (44b5506)",
136            parent=commit_id_node,
137            critical=True,  # Critical - ID must be correct
138        )
139
140        # Always perform verification - short-circuit logic will skip if
                  existence failed
141        claim = f"This ID (a github commit id) '{commit_info.commit_id or
                  'N/A'}' matches this ID '{GROUND_TRUTH['commit_id']}'"
142        await evaluator.verify(
143            claim=claim,
144            node=id_correctness,
145            sources=None,  # Simple comparison, no URL needed
146            additional_instruction="Allow minor formatting differences or
                      extra descriptions but the core 7-character commit ID
                      should exist and match exactly. Expected: 44b5506."
147        )
148
149        # Provenance check - verify the commit info is supported by
                  sources
150        provenance_check = evaluator.add_leaf(
151            id_="commit_provenance",
152            desc="Commit information is supported by provided source
                  URLs",
153            parent=commit_id_node,
154            critical=True,  # Critical - information must be substantiated
155        )
```

```python
156
157        # Always perform verification - short-circuit logic will skip if
                needed
158        # if commit_info.source_urls and commit_info.commit_id:
159        claim = f"This page shows or mentioned the github commit ID:
                '{commit_info.commit_id}'. For example, if this is exactly the
                commit page"
160        await evaluator.verify(
161            claim=claim,
162            node=provenance_check,
163            sources=commit_info.source_urls,
164        )
165

166
167   async def verify_commit_date(
168            evaluator: Evaluator,
169            parent_node: VerificationNode,
170            commit_info: CommitInfo,
171   ) -> None:
172        """
173        Verify commit date accuracy.
174        """
175        # Check if date exists
176        date_exists = evaluator.add_custom_node(
177            result=bool(commit_info.date and commit_info.date.strip()),
178            node_id="commit_date_exists",
179            description="Commit date is provided in the answer",
180            parent=parent_node,
181            critical=True,  # Critical - date is required
182        )
183
184        # Verify date correctness
185        date_correctness = evaluator.add_leaf(
186            id_="commit_date_correctness",
187            desc="Commit date matches the expected value (Dec 7, 2023)",
188            parent=parent_node,
189            critical=True,  # Critical - date must be correct
190        )
191
192        # Always perform verification - short-circuit logic will skip if
                existence failed
193        claim = f"The provided commit date '{commit_info.date or 'N/A'}'
                matches the expected date '{GROUND_TRUTH['date']}'"
194        await evaluator.verify(
195            claim=claim,
196            node=date_correctness,
197            sources=commit_info.source_urls if commit_info.source_urls
                    else None,
198            additional_instruction="Allow reasonable date format
                    variations (e.g., 'Dec 7, 2023', 'December 7, 2023',
                    '2023-12-07') but the core date should match. Expected:
                    Dec 7, 2023."
199        )
200

201
202   async def verify_authors_info(
203            evaluator: Evaluator,
204            parent_node: VerificationNode,
205            authors_info: AuthorsInfo,
206   ) -> None:
207        """
208        Verify authors information in parallel.
209        This is the second node in the sequential chain.
210        """
211        # Create parallel node for authors verification
```

```python
        authors_verification = evaluator.add_parallel(
            id_="authors_verification",
            desc="Verify all authors information in parallel",
            parent=parent_node,
            critical=False  # Non-critical to allow partial scoring
        )

        # Extract first 5 authors, pad with empty authors if needed
        provided_authors = authors_info.authors[:5] if \
            authors_info.authors else []

        # Pad with empty AuthorInfo objects for missing authors
        while len(provided_authors) < 5:
            provided_authors.append(AuthorInfo())  # Empty author info

        # Create verification nodes for each author position
        for i, author in enumerate(provided_authors):
            await verify_single_author(evaluator, authors_verification,
                author, i + 1)

async def verify_single_author(
        evaluator: Evaluator,
        parent_node: VerificationNode,
        author: AuthorInfo,
        author_number: int,
) -> None:
    """
    Verify a single author's information.
    """
    # Create parallel node for this author
    author_node = evaluator.add_parallel(
        id_=f"author_{author_number}",
        desc=f"Author {author_number} information verification",
        parent=parent_node,
        critical=False  # Non-critical to allow partial scoring
            across authors
    )

    # Check if author information exists
    author_exists = evaluator.add_custom_node(
        result=bool(author.name and author.name.strip()),
        node_id=f"author_{author_number}_exists",
        description=f"Author {author_number} name is provided",
        parent=author_node,
        critical=True  # Critical - if no name, this author slot is
            meaningless
    )

    # Verify name matches expected contributors
    name_match_node = evaluator.add_leaf(
        id_=f"author_{author_number}_name_match",
        desc=f"Author {author_number} name matches one of the
            expected contributors",
        parent=author_node,
        critical=True,  # Critical - must match an expected author
    )

    # Always perform verification - short-circuit logic will skip if
        existence failed
    expected_authors_str = ", ".join(GROUND_TRUTH['expected_authors'])
    author_name = author.real_name_from_profile if \
        author.real_name_from_profile else author.name
    claim = f"The name '{author_name or 'N/A'}' matches one of the
        names in the following list: {expected_authors_str}"
```

```
269        await evaluator.verify(
270            claim=claim,
271            node=name_match_node,
272            sources=None,  # Simple name matching
273            additional_instruction="Allow variations like 'Arthur'
                    matching 'Arthur Zucker', or reasonable name format
                    differences. Expected authors: Younes B, Arthur (or Arthur
                    Zucker), Shauray Singh, Lysandre Debut, Haotian Liu."
274        )
275
276        # Verify profile page is provided (non-critical)
277        profile_provided_node = evaluator.add_leaf(
278            id_=f"author_{author_number}_profile_provided",
279            desc=f"Author {author_number} GitHub profile page URL is
                    provided",
280            parent=author_node,
281            critical=False,  # Non-critical - nice to have but not
                    essential
282        )
283
284        profile_claim = f"This is a GitHub profile page for '{author.name
                or 'N/A'}'"
285        await evaluator.verify(
286            claim=profile_claim,
287            node=profile_provided_node,
288            sources=author.profile_url,
289        )
290
291
292    # Main evaluation entry point
293    async def evaluate_answer(
294            client,  # LLMClient type not imported, use generic annotation
295            answer: str,
296            agent_name: str,
297            answer_name: str,
298            cache: CacheClass,
299            semaphore: asyncio.Semaphore,
300            logger: logging.Logger,
301            model: str = "o4-mini"
302    ) -> Dict[str, Any]:
303        """
304        Main evaluation function for the LLaVA commit finding task.
305
306        This function implements a sequential verification strategy:
307        1. First verify commit information (ID, date, provenance)
308        2. Then verify authors information (only if commit info is
                correct)
309
310        The sequential design ensures that if commit information is wrong,
311        author verification is automatically skipped via short-circuit
                logic.
312        """
313
314        # -------- 1. Initialize evaluator --------------------------- #
315        evaluator = Evaluator()
316        root = evaluator.initialize(
317            task_id=TASK_ID,
318            strategy=AggregationStrategy.SEQUENTIAL,
319            agent_name=agent_name,
320            answer_name=answer_name,
321            client=client,
322            task_description=TASK_DESCRIPTION,
323            answer=answer,
324            global_cache=cache,
325            global_semaphore=semaphore,
```

```python
            logger=logger,
            default_model=model,
        )

        # Record ground truth information
        evaluator.add_ground_truth(GROUND_TRUTH,
            "expected_commit_and_authors_info")

        # -------- 2. Extract structured information ------------------ #

        # Extract basic commit information
        commit_info = await evaluator.extract(
            prompt=prompt_extract_commit_info(),
            template_class=CommitInfo,
            extraction_name="commit_extraction",
            source=None,  # Extract from answer text
        )

        # Extract authors information
        authors_info = await evaluator.extract(
            prompt=prompt_extract_authors_info(),
            template_class=AuthorsInfo,
            extraction_name="authors_extraction",
            source=None,  # Extract from answer text
        )

        # -------- 3. Build verification tree (Sequential) ------------- #

        # Step 1: Verify commit information (non-critical for sequential
            scoring)
        await verify_commit_info(evaluator, root, commit_info)

        # Step 2: Verify authors information (will be skipped if commit
            info fails)
        await verify_authors_info(evaluator, root, authors_info)

        # -------- 4. Return evaluation results ----------------------- #
        return evaluator.get_summary()
```

# H   Instructions for Human Annotators

## H.1   Instructions for Task Collection

---

### Task Proposal Instruction

{Introduction to this project}

**Criteria for Task Proposals**

- Tasks should:
    - Be information-seeking.
    - Be realistic, reflecting genuine scenarios or previously encountered problems.
    - Be tedious and sufficiently complex, requiring multiple intermediate steps and taking at least five minutes for a human to complete.
    - Be verifiable. The agent's response must include text and reference URLs or be verifiable through established ground truth.
    - Be single-round tasks (no user clarification required); all necessary information must be clearly included in the description.
- Tasks to avoid:
    - Tasks requiring user logins.
    - Simple or quickly resolvable tasks.
    - Tasks with global qualifiers (e.g., "cheapest," "best") that cannot be reliably verified.
    - Tasks containing vague or subjective elements (e.g., "nice restaurant").

---

### Task Refinement Instruction

We are conducting task refinement to ensure that all proposed tasks consistently meet our standards. We provide a structured checklist to help you evaluate whether each task aligns with our specified criteria. Please thoroughly go through these checks to assess, refine, or filter out the tasks accordingly.

**Checklist**

1. Realism
    a. Verify the task reflects real-world scenarios. Imagine yourself or someone you know performing this task in real life.
    b. Verify the task is not artificially combining many simple steps to increase complexity or tediousness.

    Note:
    - Certain subjectivity regarding the realism is acceptable here since people have different practical needs.

2. Clarity and Objectivity
    a. Ensure the task description is typo-free and grammatically correct.
    b. Verify that the description is clear and understandable.
    c. Ensure the task explicitly states the necessary background knowledge needed to complete the task.
    d. Make sure the task criteria are objective and unambiguous:
        - Avoid subjective terms (e.g., "nice", "good").
        - Avoid vague terms (e.g., "effective", "better") and use precise, measurable language instead.
    e. Ensure there are no alternative interpretations of the task.

3. Tediousness and Feasibility
    a. Confirm that the task takes more than 5 minutes to complete. Try performing the task, ensure it can't be quickly solved with only one or two simple searches.
    b. Confirm that the task required information can be found on publicly accessible websites where no login or paywall is required.

    Note:

---

- Familiarity with the task might cause you to underestimate the actual completion time.
- Tasks don't need to specifically challenge particular AI systems (e.g., Perplexity AI, ChatGPT Search, Deep Research).

4. Verifiability

   a. Ensure task verifiability.

      i. Draft an outline of the expected answer as a sanity check, as well as to assist future task validation. The outline should include:
         - All critical information explicitly required by the task, OR
         - Information that, while not explicitly requested, should reasonably be included in the answer based on common sense.

      ii. Ensure that the information in the outline is sufficient to verify whether an answer satisfies the following principles by using our verification and helper tools:
         - *Correctness*: The answer meets all task requirements.
         - *Source Attribution*: Each fact is supported by at least one URL source.

Verification and Helper Tools:

- *Simple LLM-as-a-Judge:*
  - Use LLMs to perform a judgment on a statement by simple logic, common-sense reasoning, or universally known facts.
- *URL-based LLM-as-a-Judge:*
  - Given a statement and URL, use an LLM to confirm whether the statement is supported by the webpage content or a screenshot.
- *Google Maps API:*
  - Given an address, retrieve the city or sub-city name (Geocoding).
  - Calculate travel time between two addresses (driving, walking, or public transit).
  - Calculate travel distance between two addresses (driving, walking, or public transit).
- *arXiv API:*
  - Search for academic papers by title and retrieve their arXiv ID.
  - Retrieve detailed information about a paper using its arXiv ID or URL.

Note:

- Be cautious with tasks involving global qualifiers (e.g., "list all", "top-k"), you must ensure such answers can be verified.
- Please be aware that our current URL-based LLM-as-a-Judge cannot obtain information from webpages that load content dynamically with additional clicks or interactions. Please avoid tasks that depend on such information for verification.
- If verification seems involving additional tools for a specific task, please discuss it with us.

5. Additional Considerations

- Tasks involving video understanding or non-English websites are currently not supported.
- Avoid tasks with rapidly changing answers (e.g., stock prices, exchange rates).
- Avoid tasks requiring extensive reasoning, complex calculations, or external tools (e.g., Python, calculators). The current focus is on information gathering via web browsing.
- If you find a task interesting but borderline according to these guidelines, discuss it with the team for further consideration.

## Task Validation Instruction

We are conducting task validation to rigorously ensure that all proposed tasks consistently meet our quality standards and are practically evaluable within our evaluation framework. During validation, please **FULLY complete each task end-to-end** by yourself, paying particular attention to potential ambiguities, overlooked edge cases, and the verifiability of all required information.

You should closely follow the structured checklist provided in the Task Refinement Instruction to evaluate realism, clarity, tediousness, and overall feasibility. Additionally, please specifically pay attention to the following validation aspects:

**Checklist Addendum for Validation**

1. **Full Completion and End-to-End Testing**

- Fully perform the task yourself from start to finish. Ensure all or most critical information can be practically located and verified from the URL sources.

2. **Feasibility of URL-based Verification**

   - Verify that each URL provided as an information source uniquely and directly supports the expected statement.
   - Avoid scenarios that are beyond our evaluation framework capabilities, such as:
     - Tasks requiring simultaneous verification from multiple distinct webpages, where the verification cannot be decomposed into independent single-page validations.
     - Tasks where the critical information is dynamically loaded, hidden, or collapsed on the webpage, making it inaccessible from the static HTML or initial page rendering. In other words, tasks that require additional webpage interactions beyond simple navigation (e.g., clicking one or multiple buttons, performing searches within the site, or scrolling to trigger dynamic loading).
     - Tasks where URLs provided are not unique or stable (e.g., search result URLs that frequently change).
     - Tasks requiring login credentials (verify this using your browser's incognito mode).

3. **Explicit Ground Truth and Evaluation Notes**

   - Clearly document the related sources and information that can be helpful for understanding the task.
   - If ground-truth information is necessary for certain criteria, note it down and carefully validate its correctness.
   - Provide explicit notes for potentially tricky evaluation scenarios or subtleties that might be easily overlooked or incorrectly handled during judge agent development.

If during validation you identify tasks that seem borderline or ambiguous with respect to our framework capabilities, promptly discuss these tasks within the team to determine their suitability or necessary adjustments.

## H.2 Instructions for Human Performance Study

We omit some examples from the instruction for clarity.

---

### Human Performance Study Instruction

**Objective**

The goal of this stage is for humans to complete tasks and provide answers, enabling us to compare human performance with AI agents. For each task, please search and browse relevant websites to gather the necessary information. Use Google Docs as a text pad for your answer as you proceed, and include the URLs of your sources as provenance for each piece of information or claim made. Each URL (webpage or PDF) should allow others to easily verify the attribution of your answer. Additionally, record videos of your completion processes for review and potential publication purposes.

**Setup Requirements**

- A clean browser context only for this purpose
  - Use a clean Chrome browser window (not incognito) without signing in, and avoid displaying sensitive personal information.
  - You can only use the browser during task completion.
- Chrome extension for timing: *Web Activity Time Tracker*
  - This extension will track the websites you visit and the time you spend on them in the browser.
  - Set *Stop the tracker if there is no action for* to *30 minutes* in the extension settings.

**Procedure for Each Task**

1. Before you begin:
   a. Understand the task clearly
      i. Carefully read the task description and ensure you clearly understand what is asked (e.g., no language barriers or a lack of domain-specific knowledge). It's okay if you do not yet know how to solve the task; planning and figuring this out are part of the task-solving.

---

      ii. Please let us know if you have prior knowledge about the assigned task (e.g., you already know the answer without searching) before starting to solve the task.

   **b.** Reset time tracker

      i. Clear previous data from the *Web Activity Time Tracker* extension. Specifically, go to `Settings → Remove all data`. This ensures tracking statistics are only for the current task from now on.

   **c.** Open a new Google Doc as a text pad for your answer

**2.** Solving the task:

   **a.** Start recording

      i. Begin screen recording before you start planning or researching for the task.

      ii. Ensure all task-related activity is recorded, avoiding external screens.

   **b.** Research and Answer

      i. Search and browse to gather accurate and reliable information. DO NOT rely on prior knowledge; actively search to verify all information.

      ii. Clearly document your response, explicitly linking each critical piece of information to its source.

         1. Our basic expectation for answers is: All critical points required by the task should be included, along with URLs to verify them (i.e., the URL where you find each information). You do not need to summarize every web page you visit.

         2. You could also write some intermediate thoughts or the reasoning process on Google Docs for yourself when completing the task, though only the critical information asked by the task is required for the final answer.

      iii. For every piece of information or statement in your answer, insert the link as attribution.

         1. Use either inline hyperlinks or numbered citations.

         2. Ensure all URLs start with `http` or `https`.

**3.** After completing the task:

   **a.** Export browsing data:

      i. Export statistics from the time tracker extension to CSV immediately

      ii. After exporting, you must stop gathering new information; only reformat your answer if you wish (e.g., if it is still cluttered). (Imagine you no longer have access to the Internet other than the answer text pad after this step.)

   **b.** Stop recording:

      i. End your screen recording promptly after exporting data.

**4.** Upload your results:

   **a.** Paste your final answer into the designated *Answer* column in the provided spreadsheet.

   **b.** Convert your recording to MP4 using *HandBrake* (preset: `Fast 1080p30`) and upload to our OneDrive folder.

   **c.** Rename the CSV as `taskID-yourName.csv`, upload it to OneDrive, and paste its URL into the spreadsheet's corresponding *Time* column.

**Notes**

- Notify us if any task takes less than 5 minutes or more than 1 hour.

- You should NOT give up on a task unless you still have not landed on a clear path to the solution after 30 minutes. However, to conserve effort, you may stop working on tasks that exceed 60 minutes, even if a solution path is evident.

- AI tools (e.g., ChatGPT) are NOT allowed.

- Personal browser extensions and setups are permitted, but keep potential video publication in mind.

**Trial Tasks**

To make sure you've understood the task and to ensure the quality of our human performance, let's start with two simple trial tasks. We will go through your answers and provide feedback. When you pass the two trial tasks, we will start assigning real tasks to you.

## H.3 Instructions for Error Analysis

We omit some examples from the instruction for clarity.

---

### Error Analysis Instruction

Please carefully read the workflow below(Figure F.1). We start by evaluating the overall correctness of the agent's entire response based on its text, and then assess the attribution of each key information.

We randomly selected answers from the selected agents, and each answer is presented to you with a Google Doc along with an annotation spreadsheet. Please follow these steps (Read, Evaluate, Comment, Collect) for error analysis:

1. Read the task and the agent's answer carefully. Make sure you fully understand the task criteria, and discuss with us when necessary.

2. Evaluate the response using the error categories detailed below.

3. Whenever you identify an error, leave a comment in the Google Doc directly on the problematic part. Your comment should briefly explain what kind of error it is and why.

4. Collect all error types you identified, and check the corresponding labels in the annotation sheet.

**Correctness Check**

This section is concerned only with the agent's full response based on the text.

1. *Incompleteness:* Our definition of Incompleteness here is limited to immediately noticeable, surface-level omissions: The agent does not provide all content explicitly requested in the task. It does not cover more subtle or long-range reasoning failures. We further define two subtypes. You may better understand our intended scope by reviewing the examples provided.

   a. *Information Not Found:* The agent explicitly states it fails, such as:
      - "I tried but failed to find...."
      - "I provide another ..., ... is not available."
   b. *Partial Completion:* The agent provides fewer items or steps than explicitly requested by the task, such as:
      - Requested 3 items but only 1 provided.
      - Asked for 5 explicit steps but only completed 3 steps.

2. *Criteria Violation:* This label is used when the agent's answer breaks explicit constraints mentioned in the task. These constraints could be things like price ranges, required formats, word limits, or instructions such as "find all." It also applies when the task comes with a ground-truth reference — for example, when the prompt clearly expects a specific answer—and the agent gives a different or incorrect one. Examples:

   - "Find an item priced under $250." → Answer provides an item of "$260".
   - "Find the specific follow-up work of a paper (the ground-truth paper is given)." → The answer gives a different, incorrect paper title.

**Attribution Check**

In this section, we examine each individual key information in the answer. Specifically, we verify 1) whether the attribution is invalid or missing, 2) whether the cited page is relevant, and 3) whether it genuinely supports the agent's claim. Specific types include:

1. *Invalid Attribution:* URL is expired, incorrectly formatted, or obviously fabricated, such as
   - The URL leads to a "page not found" error.
   - The provided arXiv link for a research paper (https://arxiv.org/abs/1234.56789) is clearly fake.

2. *Missing Attribution:* No source URL is provided for the claim. For example, in the following answer, a missing source and a valid one are presented.
   - Totokaelo
     – Address: 913 Western Avenue, Seattle, WA 98104 → Missing Attribution: no URLs (including the following product page link) can substantiate the address information.
     – Product Page: https://totokaelo.com/collections/acne-studios
   - DNA 2050

---

```
                  – Address: 700 11th Avenue, Suite 160, The Bravern, Bellevue, WA 98004 [1]
                    → Correct attribution: the link provides the address.
                  – Product Page: https://www.dna2050.com/collections/acne-studios
```

3. *Unsupported Answer:* If a reachable attribution is provided, we need to examine whether it can support the claim. If not, there are 2 subtypes of errors:

   a. *Retrieval Error:* The provided sources are irrelevant to the task, such as:
   ```
   • The task requests the list of K-pop songs in Just Dance, but the provided
     source is a discussion page without a tracklist.
   • The task requests a complete list of character abilities in Marvel Rivals,
     but the source only contains team-up abilities, unable to support the
     hallucinated ability list.
   ```

   b. *Synthesis Error:* The source contains useful information required by the task, but the answer misquotes or misinterprets it. Examples:
   ```
   • The provided product page shows a price of $220, while the answer
     incorrectly states $230.
   • The source gives a correct tracklist of songs in Just Dance, but the
     answer identifies incorrect K-pop songs.
   ```

**Notes**

- Please check our sample annotations to help you get started and use them as references during annotation.

- If you are unsure how to label an answer, please raise the issue in the group for discussion.

## H.4   Instructions for Human Evaluation

Human Evaluation Instruction

**Objective**

The goal of this study is to validate the quality and reliability of the rubrics and judge agent implementations. Your task is to independently evaluate them in two phases: rubric-level and node-level assessment.

Specifically, we have sampled 15 tasks and prepared their evaluation rubrics and evaluation results. In the rubric-level assessment phase, you will review these evaluation rubrics structured as trees, and record your feedback. In the node-level assessment phase, for tasks with rubrics agreed upon in the previous phase, you will independently score the leaf nodes of the rubric for two sampled answers per task, without reference to the judge agents' evaluation results.

**Phase 1: Rubric-Level Assessment**

For each task, you will be provided with an evaluation rubric in a tree structure. Each node in the tree includes an ID, a brief description, and its settings (e.g., sequential or parallel; critical or non-critical). Please refer to §3.3 for a detailed explanation of our rubric design.

Please read the task description carefully to understand the intended evaluation criteria, examine the rubric carefully, and use the following scale to rate each rubric:

- *Strongly Agree:* The rubric is clear, comprehensive, practical, and fully aligns with task requirements.

- *Agree with Reservations:* The rubric is generally acceptable, but minor adjustments (e.g., stricter or clearer node criteria) would enhance clarity or robustness.

- *Disagree:* The rubric is significantly flawed, impractical, or misaligned with the task criteria.

When reviewing rubrics, pay close attention to: node decomposition and ordering, prompt formulation, score aggregation strategies, etc. If the tree structure alone is not sufficient for understanding, please refer to the evaluation script to review the actual prompt wording and implementation details.

For each rubric you evaluate, you must provide a clear written justification for your rating.

Notes:

- A task may allow for multiple reasonable rubric trees. As long as the provided rubric is logically sound and aligns with the task goals, it should be accepted.

- If you disagree with any rubric, please notify us ASAP.

**Phase 2: Node-Level Assessment**

Only rubrics that passed Phase 1 will proceed to this phase. For each task, you will be given two sampled agent answers. Your goal is to independently evaluate each answer at the leaf node level, strictly following the rubric.

You will be provided with a JSON file for each task. This file contains the structured evaluation rubric tree, where all scores are unset. Each leaf node will contain a score field set to "TODO". Your task is to replace each "TODO" with a judgment:

- *1* if the answer satisfies the leaf node criterion, or

- *0* if it does not.

In other words, you are substituting your own judgment in place of the LLM-based verification in each leaf node. Please strictly follow the rubric definitions when assigning scores. If anything is unclear, refer to the evaluation scripts.

Once you have completed all annotations for a task, upload your updated JSON file with all "TODO" values replaced.

Please note that human judgment is not always perfect, especially when working with a rubric that contains hundreds of leaf nodes. To help ensure the quality of this evaluation study, we will run a script that compares your annotations with the results from the judge agent. This will generate a mismatch report highlighting all leaf nodes where your judgment differs. We will then have another annotator to validate your results and discuss with you. You should also review all mismatches, be ready to discuss with the validator and explain the rationale for your decision, and indicate whether the discrepancy may have resulted from an oversight on your part.

Notes:

- Internal node scores will be automatically calculated based on leaf evaluations; you do not need to judge them.

