# OpenReview forum: "Mind2Web 2: Evaluating Agentic Search with Agent-as-a-Judge"
_NeurIPS.cc/2025/Datasets_and_Benchmarks_Track — NeurIPS 2025 Datasets and Benchmarks Track poster_

### Official Review · Reviewer_6oUa · 2025-06-29

**Rating:** 3
**Confidence:** 4

**Summary:**

This paper tries to address the lack of evaluation benchmark datasets for deep research on long-horizon, complex questions that can have time-varying answers. As deep research is a powerful and promising paradigm to enhance current search engine, comprehensive and unbiased evaluations are required to make progress in this problem, and the paper provides a timely solution to one of many aspects of deep research. The authors asked human annotators to create many questions and then select and verify the questions to make sure the questions are hard enough for current sota agents. Then they propose an agent-as-a-judge method to create rubrics for automatic evaluation of the generated results (in texts) from various sota deep research platforms (Gemini, Grok, OpenAI Operator, etc.). The results show that the sota agents have far lower success rate or partial completion rate than human beings (who cannot solve the problem 100%), though humans take a longer time on average to complete the tasks (capped at 1 hour). Error analysis is quite valuable to find where future improvements are possible. Overall, the paper presents an important step towards more efficient and effective deep research.

**Dataset Code Accessibility:**

No

**Dataset Code Comments:**

The paper claims that only 10 questions are released, but the paper seems to provide no link to the 10 questions.

**Ethical Considerations:**

No, there are no or only very minor ethics concerns

**Final Justification:**

I have read the rebuttal and discussed them with the authors.

While I agree that collecting data can be time consuming and costly, but since this is the DB track and the significance of the dataset is of upmost value. Therefore, given the limited size of the collected dataset, I insist a rating of 3 points. Having said that, I do see the valuable parts of the paper and won't object the paper being accepted given other reviewers' high ratings. Thanks!

**Limitations Weaknesses:**

Although the paper describes how to design the rubric trees using agent and Python codes, it is not clear how the codes are run against the submitted answer text (including the provenance) as input: where should the extractor pays attention to? how to connected the extracted entities to the target entities in the user's query for verification? Is exact matching or partial matching considered a successful matching? A lot of details are not revealed in the main texts of the paper.
It would be useful to show case studies in the Appendix, and use an example in the main texts of the paper to demonstrate various steps of the algorithm (e.g., rubric trees).
The paper claims that they can handle time-varying answers, but there seems no specific component and experiment to address these sort of scenarios.
There are only 100 cases and with more resources, more cases can be collected.

**Strengths Contributions:**

Long and complex tasks that requires accessing multiple websites to find the time-varying answer is a gap in deep research, since there is no high-quality benchmark datasets. The paper is the first to tackle this gap.
The error breakdown provides insight about future direction of improvment of LLM-based deep research.
The use of rubric tree make the evaluation semi-automated and can serve as a paradigm for other similar research (though rubric used by LLM/agent are not the paper's invention).

---

> ### Author Rebuttal · Authors · 2025-07-30
>
> We sincerely thank the reviewer for recognizing our contribution to evaluating agentic search systems and for the constructive suggestions that help us further improve our paper. We acknowledge that the comments primarily focus on the clarity and details of presentation, and we will enhance these aspects in the revision. We address the key concerns as follows.
>
>
>
> ## 1. Evaluation Details
>
> > “It is not clear how the codes are run against the submitted answer text [...] where should the extractor pays attention to? [...] Is exact matching or partial matching considered successful? [...] It would be useful to show case studies in the Appendix [...] use an example in the main texts of the paper to demonstrate various steps of the algorithm (e.g., rubric trees)”
>
>
>
> We appreciate the chance to clarify our evaluation pipeline, using the IKEA task (Figures 1 & 2) as an illustrative example:
>
>
> 1. We use the Extractor module to parse the full answer. Specifically, for each furniture type requested by the task, the Extractor identifies corresponding product details provided in the answer (e.g., product name, price, color, and source URLs).
> 2. Next, for each furniture item extracted from the answer, we verify whether it meets the task-specific requirements as defined by the rubric tree shown in Figure 2 (right panel).
>    1. In this case, each verification employs the Verifier module, which dynamically compares the extracted attributes to the contents of the cited webpages. Specifically, it verifies: (a) whether the item is indeed the correct type of furniture, (b) whether the item is white, and (c) whether the listed price matches the actual price from the source webpage.
> 3. We also verify if the claimed total price falls within the specified budget ($200–$600) using a simple LLM-based verification.
> 4. The binary verification outcomes at the leaf nodes are then automatically and iteratively aggregated according to the rubric tree structure to produce a final evaluation score (Figure 2, left panel).
>
>
>
> As illustrated in this example, the Extractor takes the entire answer (which should include source URLs, otherwise attribution-related verification will fail) and parses the information specifically required for subsequent verification steps. Each extraction is accompanied by a dedicated prompt with a structured JSON object that precisely specifies where the Extractor should focus.
>
>
>
> Furthermore, since most tasks (like this IKEA example) involve time-varying information such as pricing and product availability—which frequently changes—we generally do not rely on exact or partial matches against predefined gold-standard answers. Instead, the LLM-based Verifier assesses whether each specific criterion defined in the rubric is satisfied by examining both the textual statements in the answer and the cited attribution webpages when needed.
>
>
>
> Due to page constraints, these implementation details were not fully elaborated in the main text. We currently provide the complete evaluation script for this task in Appendix G, including specific prompts for extraction and verification. General system prompts for the Extractor and Verifier are detailed in Appendix F.2. We will further clarify and highlight these points explicitly to ensure complete clarity.
>
> ----
>
> ## 2. Handling Time-Varying Answers
>
> > "The paper claims that they can handle time-varying answers, but there seems no specific component and experiment to address these sort of scenarios."
>
>
>
> The previously discussed IKEA task itself provides a straightforward example of our framework’s capability in handling time-varying answers. Our evaluation dynamically checks the correctness of statements and their supporting attributions based on cited webpages as needed, thus enabling precise verifications of time-varying answers.
>
> -----
>
> ## 3. Number of Tasks
>
> > "There are only 100 cases and with more resources, more cases can be collected."
>
> We acknowledge that expanding the task set could potentially benefit the benchmark. However, we would like to emphasize several points demonstrating the substantial value provided by the current benchmark, even with just the current 100 tasks.
>
> 1. As emphasized in lines 237-238, with fine-grained evaluation of dozens to hundreds of evaluation nodes with the partial scoring mechanism, the 100 tasks already provide great differentiation power.
> 2. While many benchmarks in the field (agentic search or broader GUI agent evaluation) often comprise hundreds of tasks, it’s worth noting that ours are more complex and have a substantially longer horizon than most existing benchmarks.
> 3. We have spent approximately 1,000 hours of human labor to construct the benchmark, which is already a non-trivial effort. The majority of the time was spent refining and validating the tasks and judge-agent scripts, to ensure the quality of our benchmark.
>
> Furthermore, the inference for the current 100 tasks is already costly. For example, the inference of the three runs of Hugging Face Open Deep Research has taken about 4,100 minutes and $2,400. Therefore, from a practical standpoint, further scaling up the task set may render the benchmark less affordable while offering only marginal benefit.

---

> > ### Comment · Reviewer_6oUa · 2025-08-06
> >
> > Regarding 2, can you specify in which table/figure that shows the experimental results of addressing time-varying answers?
> >
> > Regarding 3, I think the number of tasks is not sufficient. Given the high cost of data annotation, some semi-automated way with a powerful GPT or some sort of crowdsourcing may be a good fit (but I am not sure and the authors can provide reasons of using/not using such techniques).

---

> ### Author Response · Authors · 2025-08-09
> **Further Discussions on Time-varying Tasks and the Benchmark Size**
>
> We sincerely appreciate your follow-up questions and are happy to provide further discussions.
>
> ## 1. Additional Experiment on Time-varying Tasks
>
> It seems that the follow-up question mainly comes from explicit experiments for time-varying tasks, which is an excellent suggestion.
>
> To clarify, all tasks in our benchmark require gathering real-time information that may change over time. To address this point directly, we will add a dedicated analysis focusing on **explicitly time-varying tasks** (i.e., tasks explicitly associated with relative dates/times, such as finding music events within the next two months or identifying award-winning games from the past year; or tasks requiring information that changes frequently, such as retrieving furniture items with their current prices)
>
> We identified **44** such tasks and compared system performance on these tasks versus other tasks. As shown in the table below, most of the evaluated systems perform worse on this subset than on the remaining tasks. This is because they mainly interact with live websites through search APIs and reading HTML, instead of using Graphical User Interfaces (GUIs) like humans do. In contrast, OpenAI Operator and human participants, both excelling at interacting with live websites through GUIs, achieve relatively on-par or superior performance on time-varying tasks.
> Results (Partial Completion):
>
> | System                    | Explicitly Time-varying Tasks | Other Tasks | Δ (Explicit - Other) |
> |--------------------------|-------------------------------|-------------|-------------------------------|
> | ChatGPT Search           | 0.15| 0.24        | -0.09 |
> | Perplexity Pro Search    | 0.18 | 0.32        | -0.14|
> | OpenAI Operator          | 0.36 | 0.37        | **-0.01**  |
> | HF Open Deep Research    | 0.35  | 0.54        | -0.19 |
> | Perplexity Deep Research | 0.26| 0.37        | -0.11|
> | Grok DeepSearch          | 0.23   | 0.38        | -0.15|
> | Grok DeeperSearch        | 0.27| 0.41        | -0.14|
> | Gemini Deep Research     | 0.31 | 0.42        | -0.11|
> | OpenAI Deep Research     | 0.37 | 0.50        | -0.13  |
> | Human| 0.76  | 0.63        | **+0.13** |
>
>
> The result aligns with our intuition: agentic search systems equipped with no or only limited browsing features perform worse on time-varying tasks compared to time-invariant tasks, as they may tend to provide outdated or hallucinated information.
>
> Again, thanks for bringing out this point. We will include this analysis in the revision to strengthen the paper.
>
> ## 2. Number of Tasks and Semi-automation in Benchmark Construction
>
> While we may differ on whether the current 100 tasks represent a reasonable size, we would like to take this opportunity to further justify our choice and clarify our use of semi-automation in benchmark construction, in addition to our earlier emphasis on the benchmark’s strong differentiation power through numerous evaluation nodes, its long-horizon nature compared to existing benchmarks, and the substantial effort invested in its creation.
>
> Regarding the suggestion of using LLMs to assist with benchmark construction, we would like to clarify that **we are already using LLMs extensively in the benchmark construction process, whenever meaningful**. For example, we developed an LLM-based code-gen agent with self-debug and self-reflection to generate the initial version of the judge-agent scripts (lines 200-210, more details in Appendix F.3). That already drastically reduces the amount of human effort needed to develop such complex Agent-as-a-Judge evaluation scripts.
>
> Fundamentally, the challenge in further scaling up the number of tasks stems from the level of complexity our benchmark is aiming for. For such complex tasks, one may ask ChatGPT for some inspiration, but that would only provide limited help. To provide some evidence for that, even with our 21 dedicated and well-trained task proposers (who did use LLMs as a source of task inspiration), about two thirds of the initial tasks proposed by them were later filtered out in the refinement and validation stages due to failure to meet our complexity, realism, diversity, and verifiability standards. **Benchmark construction for such complex tasks is not bottlenecked by the lack of initial tasks (which one can get many through LLMs or crowdsourcing), but by the manual refinement and verification to ensure the tasks are high-quality enough**.
>
> Finally, because of the complexity of the tasks, conducting an evaluation on more tasks can be prohibitively expensive and time-consuming. For example, running and evaluating Hugging Face Open Deep Research on the 100 tasks already took ~2600 USD.
>
> In summary, for all the reasons listed above as well as in the previous rebuttal, we believe a benchmark size of ~100 tasks is the right size for agentic search at this stage, and we are open to further expand the benchmark as the community continues to make progress on the current benchmark.

---

### Official Review · Reviewer_z7CH · 2025-07-03

**Rating:** 5
**Confidence:** 3

**Summary:**

This paper presents **Mind2Web 2**, a benchmark designed to evaluate long-horizon web agentic systems using a novel Agent-as-a-Judge framework ([lines 69-76]). The benchmark includes 100 complex tasks, each annotated with detailed tree-structured rubrics to assess both correctness and attribution. Tasks span 6 domains and 24 sub-domains and simulate realistic user goals. These goals often require reasoning across dozens of webpages and sources ([lines 154–160], [223–225], [226–233]). The paper shows that even state-of-the-art agents achieve low success rates, highlighting the gap between current AI systems and human capabilities ([lines 83-87], [326-331]). It also proposes rigorous evaluation protocols with human-in-the-loop rubric design, metrics like Success Rate and Partial Completion ([lines 176–187]) and comparisons against human annotators ([lines 262–264]).

**Additional Feedback:**

- **Minor Clarity Improvements**
  Clarify whether the agent's use of tools or browser APIs  is permitted. This could affect how results are interpreted in an actual deployment context.

- **Error Categories Are Well Explained**
  Appendix E provides helpful qualitative examples of system errors. These support the claims around hallucination, incompleteness, and criteria violation.

- **Tree Depth and Differentiation**
  The rubrics include up to 5 layers and 267 nodes per task ([lines 223–225]). This provides significant differentiation power, even though there are only 100 total tasks ([lines 236–238]).

**Dataset Code Accessibility:**

Yes

**Dataset Code Comments:**

The dataset can be loaded easily from Kaggle and the Github repository has well organized code.

**Ethical Considerations:**

No, there are no or only very minor ethics concerns

**Final Justification:**

I have maintained my score of 5. The paper introduces a comprehensive benchmark for evaluating long-horizon web agents using a novel Agent-as-a-Judge evaluation framework and tree-structured rubrics. The rebuttal by the authors has addressed all of my concerns. The authors explained the rationale for limited rubric release to protect test set integrity and clarified that rubric abstractions could compromise the evaluation. They also confirmed the sampling procedure for the Subset-25 tasks and that the tool use is permitted. The benchmark can benefit from more rubric-level metadata, though this is a minor suggestion. Overall, the paper is technically sound and novel, and I support its acceptance.

**Limitations Weaknesses:**

- **Limited Public Evaluation Access**
 While the benchmark includes 100 tasks, only 10 include publicly available evaluation scripts ([lines 239–242]). The remaining 90 tasks are released without rubrics to prevent misuse. While this rationale is understandable, it limits reproducibility and makes it difficult to count all 100 tasks as a full open-source contribution. A possible compromise could be to release rubric metadata or abstractions to balance accessibility and integrity.

- **Unclear Subset-25 Sampling Procedure**
  The Subset-25 human evaluation is a valuable study ([lines 226–233]), but it is not very clear how the 25 tasks were selected. Clarifying the sampling procedure would improve the interpretability and trustworthiness of the human comparison.

**Strengths Contributions:**

- **Technically Sound Evaluation Framework**
  The Agent-as-a-Judge methodology tackles the challenge of evaluating open-ended, time-varying web answers by relying on known task structures and human-derived rubrics ([lines 69–76], [112–120]). This is more realistic than BrowseComp type setups that sidestep the complexity ([lines 112–120]).

- **Human-in-the-Loop Task Design**
  The benchmark construction pipeline includes reasons for choosing domains, task refinement, and multi-stage validation with human experts to ensure feasibility and clarity ([lines 154–160]). Only tasks validated by at least two annotators are included.

- **Granular Evaluation via Tree-Structured Rubrics**
  The tree-based rubric design allows more interpretable scoring. The "gate then averaging" strategy for aggregation is both intuitive and feasible for partial credit ([lines 176–187], Figure 2).

- **Empirical Findings Highlight Critical Gaps**
  The study reveals significant gaps between LLM-based systems and humans. This is specially for criteria violations, even for humans ([lines 83–87], [326–331]). This reinforces the need for structured evaluation and agent support.

- **Clear Documentation and Reproducibility**
  The example rubric script in Appendix G is easy to understand and reproduce. The authors also provide rubric examples and error analysis (Appendix E and F), increasing transparency. The authors give examples for each error category.

---

> ### Author Rebuttal · Authors · 2025-07-30
>
> We appreciate the Reviewer’s highly positive and constructive feedback. Now we clarify the points raised in the Limitations Weaknesses and Additional Feedback.
>
> ## 1. Public Evaluation Access
>
> > "While the benchmark includes 100 tasks, only 10 include publicly available evaluation scripts ([lines 239–242]). The remaining 90 tasks are released without rubrics to prevent misuse. While this rationale is understandable, it limits reproducibility and makes it difficult to count all 100 tasks as a full open-source contribution. A possible compromise could be to release rubric metadata or abstractions to balance accessibility and integrity."
>
> Thank you for acknowledging our rationale behind releasing evaluation scripts only for the public development set (10 tasks).
>
> We have carefully considered your suggestion of providing "rubric metadata or abstractions." While we recognize the value of enhancing transparency and reproducibility, releasing detailed rubric abstractions carries significant risks. Specifically, with our extensively optimized codebase, having such rubric abstractions might enable automatic or semi-automatic reproduction of the full judge-agent scripts for the test set, since the primary intellectual effort in judge-agent development lies exactly in designing comprehensive and rigorous rubrics that accurately reflect each task's criteria. Consequently, releasing such abstractions could seriously undermine our goal of protecting the integrity of the test set.
>
> Furthermore, the task descriptions themselves contain sufficient information to infer the rubric, thus, the release of rubric abstraction might be unnecessary. For example, the IKEA furniture purchasing task naturally involves verifying each item's requirements (e.g., confirming the color white) and validating that the provided information precisely matches cited webpage sources.
>
> The publicly released scripts for the 10 tasks primarily aim to concretely illustrate our evaluation framework and to facilitate immediate use for developmental and research purposes. For the remaining private tasks, we will evaluate submitted answers ourselves upon leaderboard submission, and we will bear all associated evaluation costs.
>
> Additionally, we believe that the full release of our refined and optimized codebase is itself a substantial contribution, enabling the community to easily adapt, scale, or further develop our evaluation framework for various scenarios.
>
> Finally, we hope to stress that the curation of tasks itself is highly non-trivial, as it requires substantial human effort in their proposal, refinement, and validation to ensure task diversity, complexity, realism, and verifiability. The complete set of 100 tasks will be a valuable resource to the community in itself.
>
> ______________
>
>
> ## 2. Unclear Subset-25 Sampling Procedure
>
> > "The Subset-25 human evaluation is a valuable study ([lines 226–233]), but it is not very clear how the 25 tasks were selected. Clarifying the sampling procedure would improve the interpretability and trustworthiness of the human comparison.""
>
> We thank the reviewer for pointing this out. The tasks in Subset-25 were randomly selected from the full set of tasks. We will make it clearer in the revised version.
>
> ______________
>
>
> ## 3. Clarification on Whether Agent Use of Tools or Browser APIs is Permitted
>
> > "Clarify whether the agent's use of tools or browser APIs is permitted. This could affect how results are interpreted in an actual deployment context."
>
> Yes, our evaluation allows all types of agents, imposing no constraints on their use of external tools or browser APIs (as mentioned in lines 91, 100). Our current experiments already include a diverse range of systems, such as search-augmented LLMs, browser-based web agents, and Deep Research systems (some of which utilize advanced tools like Python interpreters and text-only or multimodal web browsers).

---

> ### Comment · Reviewer_z7CH · 2025-08-05
>
> Thank you for thoroughly addressing my concerns. I appreciate the clarifications regarding all 3 points. Your rationale for limiting access to full evaluation scripts for the remaining 90 tasks is understandable, especially to maintain the integrity of the benchmark. I would recommend making it more explicit in the appendix/paper which 10 tasks have been open-sourced and where the readers can find them. This would improve transparency and help readers quickly identify the scope of the public development set without any confusion.

---

> ### Author Response · Authors · 2025-08-06
> **Thank you**
>
> We appreciate Reviewer z7CH’s understanding of our rationales about public evaluation access, and thank them for confirming that their concerns have been thoroughly addressed. We also appreciate the suggestion to include explicit references for easier access to the resources, and we will make sure to include these into the final revision.

---

### Official Review · Reviewer_TYGS · 2025-07-07

**Rating:** 5
**Confidence:** 4

**Summary:**

The paper introduces *Mind2Web 2*, a benchmark designed to evaluate deep agentic search systems on realistic, long-horizon web-based tasks that require real-time browsing and multi-source information synthesis. To address the challenges of evaluating the complex, time-varying answers, the authors propose an Agent-as-a-Judge framework, which uses task-specific, tree-structured rubrics and LLM-based judge agents to assess both factual correctness and source attribution. Evaluation results on frontier agentic search systems reveal a large headroom for improvements.

**Dataset Code Accessibility:**

Yes

**Ethical Considerations:**

No, there are no or only very minor ethics concerns

**Final Justification:**

I appreciate the additional experiments on Llama-4-scout, which help support the usability of the proposed Agent-as-a-Judge framework. I’m also glad to see that important details and analyses (many are currently in the appendix) will be incorporated and better organized in the main text, which I believe will significantly benefit readers. Overall, my concerns have been sufficiently addressed; accordingly, I’ve updated my score from 4 to 5.

**Limitations Weaknesses:**

- The paper would benefit from a more detailed description of the dataset construction process, particularly in ```Section 3.2```, which currently appears overly simplified and omits potentially important details. Given that this is a dataset-focused contribution, this section should be expanded and more thoroughly elaborated. For example: what are the criteria for selecting “*realistic*”, “*long-horizon*”, and “*laborious*” tasks; what are the annotation guidelines for task proposers, refinement experts, and validation experts; how many annotators are there in each group. While some supporting statistics (*e.g.*, human completion time) are provided, they are not clearly linked to the task design or selection methodology. Additionally, many relevant  details seem to be relegated to the Appendix without explicit references in the main text. These omissions increase the challenge for readers to fully understand the construction and rigor of the proposed dataset.

- The proposed rubric-based Agent-as-a-Judge design involves substantial complexity that may limit the broader applicability and robustness of the framework. Given the heavy reliance on multiple agent components, custom scripts, and tool integrations, the evaluation process could be difficult to replicate or adopt in academic settings, raising concerns about long-term accessibility and usability. For example, will the evaluation procedure remain compatible with more affordable models as alternatives to o4-mini? Or when o4-mini is deprecated/updated? The approach also appears to be tightly coupled to each individual task, with dozens to hundreds of evaluation nodes and dedicated judge agents per task. Given the high engineering effort required per task, the scalability of the benchmark appears limited. Collectively, these factors raise concerns about the sustainability of the proposed evaluation methodology.

- Is there any meta-evaluation conducted to support the accuracy of the proposed Agent-as-a-Judge framework? This is particularly important given that humans achieve only around a 0.3 success rate under this evaluation, which seems surprisingly low. Although the abstract mentioned (```Line 13```) high agreement with human judgments, I could not find supporting evidence for this claim in the main paper (If I missed any such analysis, please kindly point it out).

**Strengths Contributions:**

- Mind2Web 2 introduces a valuable and realistic benchmark for assessing long-horizon, time-sensitive web tasks, which are increasingly relevant for emerging agent systems -- making this work a timely addition to the field.
- The accompanying error analysis is thorough and insightful, offering a better understanding of common failure modes and providing useful guidance for future research directions and system diagnosis.
- Overall, the paper is well-written, clearly structured, and easy to follow. The presentation, motivation, and contributions of the work are effectively communicated.

---

> ### Author Rebuttal · Authors · 2025-07-31
>
> ## 1. Task Construction Details
>
>
> We appreciate the reviewer’s suggestion to elaborate further on the dataset construction process. We agree that, while we have provided some details in Appendix D, better integration of these into the main text would enhance clarity. We will revise our main text accordingly, explicitly referencing key details from the appendix to ensure transparency and ease of understanding.
>
> To provide further details, we will provide the following statistical breakdown of each stage below in the revision:
>
> - **Proposal Stage (21 task proposers)**: 330 tasks were initially proposed (excluding incomplete or clearly unsuitable tasks considered random drafts).
> - **Refinement Stage (7 refinement experts, closely collaborating with original proposers)**: 115 tasks were carefully filtered and refined according to our core design principles (Realism, Tediousness, Clarity & Objectivity, and Verifiability).
> - **Validation Stage (4 validation experts)**: 100 final tasks were rigorously validated by at least two independent validators, ensuring clarity, feasibility, and complete verifiability through comprehensive manual execution and verification. Of the 15 tasks filtered out during this stage, 2 were excluded solely because we had already reached our target number of 100 tasks. The majority of the remaining excluded tasks were removed due to subtle verification issues identified during detailed manual assessments, and a few were eliminated because of substantial overlap with existing tasks. In the final set of 100 validated tasks, two tasks underwent minor modifications to resolve slight ambiguities identified during the subsequent answer-based judge-agent script validation.
>
> Furthermore, we will also provide the raw annotation guidelines for each stage in the revised appendix to help futher illustrate the details of the process.
>
> -----
>
> ## 2. Complexity, Scalability, and Long-term Usability of the Agent-as-a-Judge Framework
>
>
> We could understand the concerns regarding the complexity of our evaluation framework, as well as the complexity of building the judge agents. We would like to emphasize and clarify the following points to address the concerns:
>
>
> 1. Necessity of this complexity: The complexity of evaluation directly stems from the inherent difficulty of thoroughly evaluating realistic, long-horizon and complex tasks that have complex, time-varying and multi-faceted answers (often containing hundreds to thousands of words and dozens of source links). Even human annotators find it challenging and time-consuming to comprehensively verify every criterion (during our manual error analysis). Existing benchmarks in the field have typically sidestepped these evaluation complexities. However, as agentic search systems become increasingly capable, it is crucial that benchmarks do not compromise task realism and complexity merely for evaluation convenience.
>
>
> 2. Complexity of the framework:  While the components might appear to be complex, our judge agents primarily rely on just two LLM-based components: Extractor and Verifier (Section 3.4), which are sufficient for most of the tasks. Only 7 tasks required Google Map API as an additional tool for certain minor parts within their evaluation. The core framework remains relatively compact and easy to manage.
>
>
> 3. Scalability and sustainability: Despite the dozens or hundreds of evaluation nodes in each judge-agent scripts, their development has been significantly streamlined by: (1) a unified tree-structured rubric that provides unified evaluation logic, (2) an optimized, modular codebase that contains reusable and well-defined components (Extractor and Verifier), rubric tree management, and automatic score aggregation, (3) an automated code generation workflow, which incorporates self-debugging and self-reflection mechanisms. These have dramatically reduced manual effort for script creation and maintainance relative to manual scripting from sratch. While human effort is still required to review and refine the scripts initially generated, this process has been much more lightweighted, and mainly involves adjustment of the rubric tree structure and specific prompts for certain extractions and verifications based on an initial version automatically generated.
>
>
>
>
>
>
> > will the evaluation procedure remain compatible with more affordable models as alternatives to o4-mini? Or when o4-mini is deprecated/updated?
>
> We thank the reviewer for raising this important point of model compatibility. To demonstrate that our framework is not restricted solely to a single model (o4-mini), we have conducted a new meta-evaluation with the open-source Llama 4 Scout model (selected due to its native structure JSON output support as well as its multimodal capabilities, which are two main requirements for the evaluation base model), in addition the existing one for o4-mini:
>
> | Model Name    | Discrepancies | Accuracy |
> | ------------- | ------------- | -------- |
> | o4-mini       | 2/287         | 99.30%   |
> | llama-4-scout | 20/287        | 93.03%   |
>
>
> Although Llama 4 Scout shows lower accuracy compared to o4-mini, its overall accuracy remains high (>90%), which is still higher than recently reported automatic evaluation in simpler web navigation tasks ([1]) (typically below 90%). We attribute this success to our tree-structured rubric design that cleanly decomposes the complex evaluation into straightforward binary verifications, the agentic code generation pipeline for generating judge agents, as well as the rigorous human refinement process. These collectively make the evaluation robust enough to yield reliable results even when using more affordable, open-source models. We will include this ablation study in the revision and will extend it with larger llama models (e.g., LLama 4 Maverick) and other open-source MLLMs that support structure output.
>
>
> [1] Xue T, Qi W, Shi T, et al. An illusion of progress? assessing the current state of web agents. COLM 2025
>
>
> -----
>
> ## 3. Meta-Evaluation and Low Human Scores
>
>
>
>
> > "Is there any meta-evaluation conducted to support the accuracy of the proposed Agent-as-a-Judge framework? Although the abstract mentioned (Line 13) high agreement with human judgments, I could not find supporting evidence for this claim in the main paper (If I missed any such analysis, please kindly point it out)."
>
>
> We did conduct a human evaluation of the Agent-as-a-Judge framework, detailed in Appendix F.5. In brief, on a randomly sampled set of 10 tasks, the accuracy of the judge agents reaches more than 99%. Given its critical importance, we plan to reorganize the main text in the revision so that we can move this human evaluation from the appendix into the main body of the paper.
>
>
> > "humans achieve only around a 0.3 success rate under this evaluation, which seems surprisingly low"
>
>
> The seemingly low human success rate arises because humans frequently experience cognitive fatigue and have limited working memory when faced with prolonged, detail-intensive tasks. For instance, in a task that requires retrieval of news articles with nuanced constraints, all the human participants exhibit various forms of oversight or carelessness regarding subtle details or overall task requirements, resulting in task failures. In contrast, most agent systems accurately interpret the task and articles and achieve better performance.
>
>
> This underscores an important point we wish to convey: *agentic search has substantial potential to augment human cognition by automating away the legwork and allowing us to focus our limited cognitive capacity on things that matter more, such as critical decisions and active oversight*.
>
>
> Additionally, we have labeled the errors in human answers (as illustrated in Fig. 3), we will later include the above analysis, along with more human error examples and analysis in the revision.

---

> > ### Comment · Reviewer_TYGS · 2025-08-06
> >
> > Thank you to the authors for their detailed responses, which address most of my concerns. In particular, I appreciate the additional experiments on Llama-4-scout. I’m also glad to see that many important details and analyses will be incorporated and organized into the main text, which I believe will significantly benefit readers. Overall, my concerns have been sufficiently addressed, accordingly, I’ve updated my score from 4 to 5.

---

> > > ### Author Response · Authors · 2025-08-09
> > > **Thank you**
> > >
> > > Thank you very much for taking the time to review our additional experiments and discussions. Your constructive feedback has helped us improve the work, and we truly appreciate your recognition of our efforts!

---

### Official Review · Reviewer_fgmq · 2025-07-07

**Rating:** 5
**Confidence:** 4

**Summary:**

The authors propose a new dataset of 100 tasks to evaluate deep research systems. Each task is very time-consuming for users (up to an hour) and requires to visit dozens of websites and hundreds of webpages to get the answer. The work contains short description of the dataset, evaluation techniques based on Rubric Trees. The authors conduct experiments for ChatGPT Search, Perplexity Pro Search, OpenAI Operator, HF Open Deep Research, Perplexity Deep Research, Grok DeepSearch, Grok DeeperSearch, Gemini Deep Research, OpenAI Deep Research.

**Additional Feedback:**

-	Lines 145-151: It would be nice to know the funnel of tasks. In particular, statistics:

o	the number of them at proposal stage

o	the number of them at refinement stage

o	the number of them at validation stage

o	the number of the final dataset ( = 100 )


-	Line 159 “determinism” You consider deterministic tasks (and it is clear why), but the general population (distribution) of user queries is far from that. How should we process and take into account non-deterministic tasks? At least a thorough discussion on this matter would be valuable.


-	Section 3.3: Rubric Tree. How are these rubrics used? Are they used at evaluation stage only? Do search engines (deep research engines etc.. ) have access to them?


-	Section 3.3: Rubric Tree. How are these rubrics used? (given Lines 200-210) Are they constructed purely by experts or by LLMs?


-	Line 191 “the provenance” What is it exactly? That in Fig 1? Does it include URLs only or web page with their full content?


-	Table 1 in title: “average number of required actions per tasks” How does it calculated ? By experts or average run of search engines or just imported from corresponding literature? If the latter, do we have inconsistency in measuring methodology?


-	Line 217: “distribution across domains is shown in Figure 1” I don’t see distribution in Figure 1. Do you mean another Figure?


-	Lines 313 and 322. Is it possible that the same answer has both Criteria Violation and  Synthesis Error? I.e., should a subcategory be unique for a single answer?




Minor

-	Table 2: the order of columns in tables (a) and (b) is not consistent. It would be nice to swap “Min” and “Max” columns. In table (b) to make easier reading.

**Dataset Code Accessibility:**

Partly

**Dataset Code Comments:**

Only 10% are available, the other part is hidden to conduct evaluation of systems and creating leaderboard

**Ethical Considerations:**

No, there are no or only very minor ethics concerns

**Limitations Weaknesses:**

Presentation issues (see questions in Additional Feedback)


More statistics on dataset collection phase  (see questions in Additional Feedback)


Very small public part available (10 tasks ! = 10% of 100 )

**Strengths Contributions:**

Overall, very good work, that extends frontiers in evaluation deep research tools and systems.


Strengths:

-	Accessible reading

-	Novel dataset with characteristics that outperforms related ones

-	Currently very actual

---

> ### Author Rebuttal · Authors · 2025-07-31
>
> We appreciate the reviewer’s positive feedback and constructive suggestions. Below we address the concerns and questions raised.
>
> ## 1. Limited number of tasks in public set
>
> > (**Limitations Weaknesses**) Very small public part available (10 tasks ! = 10% of 100 ); (**Dataset Code Comments**) Only 10% are available, the other part is hidden to conduct evaluation of systems and creating leaderboard
>
> We would like to clarify that we have released all 100 tasks (as mentioned in lines 239-242; available at the “Dataset URL” on Openreview). Additionally, we provide full evaluation scripts for the public set, primarily aiming to concretely illustrate our evaluation framework and to facilitate immediate use for developmental and research purposes. For the private test set (90 tasks), we publicly release all task descriptions but intentionally withhold the evaluation scripts to prevent contamination or misuse (e.g., misused as RL rewards).
>
> ----
>
> ## 2. Detailed statistics on dataset collection stages
>
> We appreciate the suggestion of providing detailed statistics about our dataset curation pipeline. Below is the comprehensive breakdown that we will include in the revision:
>
> * **Proposal Stage (21 task proposers involved)**:
>   330 tasks were initially proposed (excluding incomplete or clearly unsuitable tasks considered as random drafts).
>
> * **Refinement Stage (7 refinement experts collaborating closely with original proposers)**:
>   115 tasks were carefully filtered and refined according to our core design principles (Realism, Tediousness, Clarity & Objectivity, and Verifiability).
>
> * **Validation Stage (4 validation experts)**:
>   100 final tasks were rigorously validated by at least two independent validators, ensuring clarity, feasibility, and complete verifiability through comprehensive manual execution and verification. Of the 15 tasks filtered out during this stage, 2 were excluded solely because we had already reached our target number of 100 tasks. The majority of the remaining excluded tasks were removed due to subtle verification issues identified during detailed manual assessments, and a few were eliminated because of substantial overlap with existing tasks.
>
> In the final set of 100 validated tasks, two tasks underwent minor modifications to resolve slight ambiguities identified during the subsequent answer-based judge-agent script validation.
>
> We will also provide the raw instructions given to human annotators at each stage to better illustrate the task collection process in the revision.
>
> ----
>
> ## 3. Clarification on determinism (Line 159)
>
> We appreciate the opportunity to clarify the nature of our task and evaluation.
>
> The word "determinism" may have led to some confusion. To clarify, we do not require tasks to have fixed or deterministic answers. On the contrary, the tasks are allowed and highly encouraged to be dynamic in nature and have answers that vary over time.
>
> The previous use of "determinism" was intended to convey the determinism of the evaluation criteria, meaning that each task is designed so that the evaluation process itself (whether an answer satisfies the task’s requirements) is unambiguous and can be deterministically evaluated by a pre-defined task specific rubric, regardless of variability in the answers. For example, in the IKEA task in Figure 1, we carefully ensure that all required criteria can be determined unambiguously (e.g., identifying whether furniture is white is deterministic, whereas assessing whether furniture is good-looking is subjective and thus avoided), and confirm that eligible items always exist (or always nonexist). We will revise that sentence at line 159 to avoid future confusion.
>
> -----
>
> ## 4. On rubric usage and rubric tree construction
>
> We clarify that agent systems being tested only receive task descriptions and have no access to rubric trees.
>
> Each task has its own dedicated rubric tree specifically designed to reflect evaluation criteria following the unified framework described in Section 3.3. In practice, each rubric tree is implemented as a judge-agent Python script, incorporating workflows and extraction logic to facilitate the rubric-style evaluation (an example is provided in Appendix G). Initially, these scripts are automatically generated using an LLM, and subsequently refined through automated debugging and rigorous expert validation (see Appendix F.3-F.4 for details).
>
> We will revise the wording in Section 3.4 to avoid confusion.
>
> ----
>
> ## 5. Clarification on "Provenance" in answers (Line 191)
>
> By "provenance," we mean the **URL-based source attribution** provided by the agent within their answer. For example, many agents always have a “sources” section at the end of their answer, with dozens of source links. During evaluation, these URLs are automatically extracted by our Extractor module as needed for subsequent verification, and both the textual content and screenshots of the webpages corresponding to the extracted URLs are utilized.
>
> We will consistently use the phrase "source attribution/links" throughout the revision to avoid potential confusion from varying terminology.
>
>
>
> -----
>
> ## 6. Calculation of "average number of required actions per task" (Table 1)
>
> By "average number of required actions per task," we mean the average number of steps or actions typically required to complete the tasks (e.g., clicks for browser-based agents, API-calls, or visited webpages for search-based agents). We use the average action-step numbers from the original papers whenever explicitly reported or estimated, or there are annotation available to calculate. While minor inconsistencies may exist due to differing methodologies or agent action spaces, we ensure these reported numbers reliably reflect the relative task horizon of each benchmark.
>
> ----
>
> ## 7. Domain distribution in Figure 1 (Line 217)
>
> The mention of "distribution across domains" refers to the central section labeled "Diverse Domains" in Figure 1, indicating task proportions across 6 broad domains. A more detailed domain distribution figure is provided explicitly in Appendix D.1, covering all 100 tasks across 6 broad domains and 24 sub-domains.
>
> ------
>
> ## 8. Clarification regarding error categories
>
> Error subcategories are not mutually exclusive. A single answer can simultaneously exhibit multiple types of errors. We will explicitly state in our revised text that multiple errors can occur within the same answer.
>
> ------
>
> ## 9. Minor presentation issue (Table 2)
>
>
> In the revised version, we will reorder the columns for consistency.
>
> ------
>
> Again, we thank Reviewer fgmq for the thorough and insightful review, which has allowed us to clarify and enhance the presentation of our work.

---

> > ### Comment · Reviewer_fgmq · 2025-08-07
> > **Thank you for your clarification**
> >
> > Thank you for your clarification and answers.
> > Well done

---

> > > ### Author Response · Authors · 2025-08-09
> > > **Thank you**
> > >
> > > Thank you very much for taking the time to review our rebuttal and discussions. We truly appreciate your recognition of our work and your encouraging feedback!

---

### Note · Authors · 2025-08-16

We thank all reviewers for their time and thoughtful engagement throughout the review and discussion phases.

We appreciate the strong consensus on several key aspects of our work, including the novelty, timeliness, and realism of the benchmark (fgmq, TYGS, 6oUa), the practicality and value of our Agent-as-a-Judge evaluation framework (z7CH, 6oUa), the thorough error analysis (TYGS, z7CH, 6oUa), and the clarity and accessibility of our writing (fgmq, TYGS, z7CH).

We have carefully addressed each of the questions and concerns raised, and are glad to see that most concerns are addressed through our rebuttal. While reviewer 6oUa remains concerned about the benchmark size and additional evidence on time-varying tasks, we believe our additional analyses and discussions have addressed these points. We respectfully differ in the interpretation of our benchmark scale, for which we have provided thorough justification, and we remain open to future expansion as the community continues to progress.

We will revise the paper accordingly to further improve clarity and readability, and will include the following key additions:

- **Complete statistics** for each dataset collection stage (numbers of tasks and annotators), with the full annotation guidelines (fgmq, TYGS)
- An additional **ablation study of the judge-agent base model**, demonstrating the robustness of our evaluation across models (e.g., Llama-4-Scout) (TYGS)
- An additional experiment on **explicitly time-varying tasks**, highlighting performance gaps in most evaluated systems (6oUa)

Again, we thank all reviewers for their constructive feedback, which has significantly improved the quality and clarity of our work.

Best regards,
Authors of Mind2Web 2

---

### Decision · Program_Chairs · 2025-09-18

**Decision:**

Accept (poster)

**Comment:**

The paper examines the ability of agents for long-horizon tasks that require real-time web interaction and extensive information synthesis. A benchmark of 100 tasks is constructed. Several agentic search systems are used to test their abilities for the tasks. A novel agent-as-a-judge framework is proposed to automatically evaluate the correctness and source attribution of answers, which rely on tree-structured rubrics and rubric-based judge agents. The evaluation with the agentic search systems shows that the current systems have limited ability for the tasks. Interesting error analysis reveals the problems to address in the future.

Strength: Overall, the paper proposes an interesting benchmark. The proposed agent-as-a-judge framework is new and interesting.

Weakness: The reviewers have mentioned a few weaknesses such as only 10 of the 100 tasks are available, the complexity of the method, etc.

Discussion: The authors provided detailed rebuttals to the reviewers' comments. The perceived weaknesses are explained to the satisfaction of the reviewers. Some reservation remains (reviewer 6oUa) due to the lack of some technical details and the limited size of the dataset.

Considering all the reviews, this paper (and the benchmark) provides a valuable contribution to the research community. It is recommended to accept it.

===== FINAL UPDATE FROM DB Track PCs ====

The final decision for this paper has been taken by the program chairs after consultation with the SACs. All Senior Area Chairs have ranked papers according to the feedback from the AC during the review process. We decided to leave the original meta-review to reflect the opinion of the AC in light of the initial discussions with reviewers and SAC.